# Towards Last-layer Retraining for Group Robustness with Fewer Annotations

**Tyler LaBonte**[1]     **Vidya Muthukumar**[2,1]     **Abhishek Kumar**[3]

[1]H. Milton Stewart School of Industrial and Systems Engineering, Georgia Institute of Technology
[2]School of Electrical and Computer Engineering, Georgia Institute of Technology
[3]Google DeepMind
{tlabonte, vmuthukumar8}@gatech.edu    abhishk@google.com

## Abstract

Empirical risk minimization (ERM) of neural networks is prone to over-reliance on spurious correlations and poor generalization on minority groups. The recent *deep feature reweighting* (DFR) technique [33] achieves state-of-the-art group robustness via simple last-layer retraining, but it requires held-out group and class annotations to construct a group-balanced reweighting dataset. In this work, we examine this impractical requirement and find that last-layer retraining can be surprisingly effective with no group annotations (other than for model selection) and only a handful of class annotations. We first show that last-layer retraining can greatly improve worst-group accuracy even when the reweighting dataset has only a small proportion of worst-group data. This implies a "free lunch" where holding out a subset of training data to retrain the last layer can substantially outperform ERM on the entire dataset with no additional data or annotations. To further improve group robustness, we introduce a lightweight method called *selective last-layer fine-tuning* (SELF), which constructs the reweighting dataset using misclassifications or disagreements. Our empirical and theoretical results present the first evidence that model disagreement upsamples worst-group data, enabling SELF to nearly match DFR on four well-established benchmarks across vision and language tasks with no group annotations and less than $3\%$ of the held-out class annotations. Our code is available at https://github.com/tmlabonte/last-layer-retraining.

## 1   Introduction

Classification tasks in machine learning often suffer from *spurious correlations*: patterns which are predictive of the target class in the training dataset but irrelevant to the true classification function. These spurious correlations, often in conjunction with the target class, create *minority groups* which are underrepresented in the training dataset. For example, in the task of classifying cows and camels, the training dataset may be biased so that a desert background is spuriously correlated with the camel class, creating a minority group of camels on grass backgrounds [5]. Beyond this simple scenario, spurious correlations have been observed in high-consequence applications including medicine [79], justice [9], and facial recognition [42].

Neural networks trained via the standard procedure of empirical risk minimization (ERM) [68], which minimizes the average training loss, tend to overfit to spurious correlations and generalize poorly on minority groups [18]. Even worse, it is possible for ERM models to rely exclusively on the spurious feature and incur minority group performance that is no better than random guessing [60]. Therefore, maximizing the model's *group robustness*, quantified by its worst accuracy on any group, is a desirable objective in the presence of spurious correlations [58].

37th Conference on Neural Information Processing Systems (NeurIPS 2023).

In contrast to more generic distribution shift settings (*e.g.,* domain generalization [35]), the presence of spurious correlations enables the improvement of group robustness merely by addressing model bias (without collecting additional minority group data). The recently proposed *deep feature reweighting* (DFR) [33, 28] technique efficiently corrects model bias by retraining the last layer of the neural network, a simple procedure which achieves state-of-the-art group robustness. The key hypothesis underlying DFR is that ERM models which overfit to spurious correlations still learn *core features* that correlate with the ground-truth label on all groups, but they perform poorly because they overweight the spurious features in the last layer. Ostensibly, retraining the last layer on a group-balanced *reweighting dataset* would then upweight the core features and improve worst-group accuracy.

DFR compares favorably to existing methods such as *group distributionally robust optimization* (DRO) [58], which requires group annotations for the entire training dataset. However, DFR still necessitates a smaller reweighting dataset with group and class annotations to achieve maximal performance [33]. This requirement limits its practical application, as the groups are often unknown ahead of time or difficult to annotate (*e.g.,* due to financial, privacy, or fairness concerns).

**Our contributions**

In this paper, we present a comprehensive examination of the performance of last-layer retraining in the absence of group and class annotations on four well-established benchmarks for group robustness across vision and language tasks.[1] We first investigate the necessity of the reweighting dataset being balanced across groups, and we show that last-layer retraining can substantially improve worst-group accuracy *even when the reweighting dataset has only a small proportion of worst-group data*. Based on this observation, we propose ***class-balanced last-layer retraining*** as a simple but strong baseline for group robustness without group annotations. We show that, on average over the four datasets, this method achieves $94\%$ of DFR worst-group accuracy compared to $76\%$ without class balancing.

The strong performance of class-balanced last-layer retraining reveals a "free lunch" with practical ramifications. While class-balanced ERM was recently proposed by [27] as a competitive baseline for worst-group accuracy, we show that instead of using the entire training dataset for ERM, dependence on spurious correlations can be reduced by randomly splitting the training data in two, then performing ERM training on the first split and class-balanced last-layer retraining on the second. Our experiments indicate that this technique can improve worst-group accuracy by up to $17\%$ over class-balanced ERM on the original dataset using *no additional data or annotations (even for model selection)* – a surprising and unexplained result given that the two splits have equally drastic group imbalance.

While retraining the last layer on a class-balanced held-out dataset can be effective, it is inferior to DFR when group imbalance is large. To close this gap, we propose ***selective last-layer finetuning*** (SELF), which selects a small, more group-balanced reweighting dataset and finetunes the last layer instead of retraining. We implement SELF using points that are either *misclassified* or *disagree* in their predictions relative to a regularized model. Disagreement SELF does not require class annotations for the entire held-out set (only for the disagreements). We show that *disagreement SELF between the ERM and early-stopped models* performs the best in general, surpassing class-balanced last-layer retraining by up to $12\%$ worst-group accuracy with less than $3\%$ of the held-out class annotations.

Overall, our work shows benefits of last-layer retraining well beyond a group-balanced held-out dataset. Our results call for *further investigation of the DFR hypothesis*: while group balance is the most important factor in DFR performance, we show that a significant gain is solely due to class balancing, and the performance discrepancy between misclassification SELF and disagreement SELF suggests that worst-group accuracy may be affected by characteristics of the reweighting dataset other than group balance. Our main results are summarized and compared to previous methods in Table 1.

## 2 Related work

This work subsumes and improves our two previous workshop papers: [38] broadly covers Section 4, while [37] represents preliminary investigation into Section 5, which we have substantially updated with new methodology, evaluation, and theoretical analysis for this version.

---

[1]Our methods in Section 4 do not require group annotations at all, while our SELF method in Section 5 assumes access to a small validation set with group annotations for model selection following previous work [58, 41, 48, 33, 28]. We show in Appendix B that SELF is robust even with $1\%$ of these annotations.

Table 1: **Comparison to other group robustness methods.** We discuss our DFR implementation in Section 3, and our proposed methods of class-balanced (CB) last-layer retraining and early-stop (ES) disagreement SELF in Sections 4 and 5, respectively. Class-balanced ERM is trained on the combined training and held-out datasets. DFR uses group annotations on the held-out dataset, while Group DRO-ES requires them for the training dataset. ES disagreement SELF uses class annotations on the training dataset (for ERM), but *requests as few as 20 labels* from the held-out dataset. All methods except ERM and CB last-layer retraining use a small set of group annotations for model selection. We list the mean and standard deviation over three independent runs.

| Method | Annotations | | Worst-group test accuracy | | | |
|---|---|---|---|---|---|---|
| | Group | Class | Waterbirds | CelebA | CivilComments | MultiNLI |
| Class-balanced ERM | ✗ | ✓ | $81.9_{\pm 3.4}$ | $67.2_{\pm 5.6}$ | $61.4_{\pm 0.7}$ | $69.2_{\pm 1.6}$ |
| JTT [41, 27] | ✗ | ✓ | $85.6_{\pm 0.2}$ | $75.6_{\pm 7.7}$ | – | $67.5_{\pm 1.9}$ |
| RWY-ES [27, 28] | ✗ | ✓ | $74.5_{\pm 0.0}$ | $76.8_{\pm 7.7}$ | $78.9_{\pm 1.0}$ | $68.0_{\pm 0.4}$ |
| CnC [81] | ✗ | ✓ | $88.5_{\pm 0.3}$ | $\mathbf{88.8_{\pm 0.9}}$ | – | – |
| CB last-layer retraining | ✗ | ✓ | $92.6_{\pm 0.8}$ | $73.7_{\pm 2.8}$ | $\mathbf{80.4_{\pm 0.8}}$ | $64.7_{\pm 1.1}$ |
| ES disagreement SELF | ✗ | ✗ | $\mathbf{93.0_{\pm 0.3}}$ | $83.9_{\pm 0.9}$ | $79.1_{\pm 2.1}$ | $\mathbf{70.7_{\pm 2.5}}$ |
| DFR (our impl.) | ✓ | ✓ | $92.4_{\pm 0.9}$ | $87.0_{\pm 1.1}$ | $81.8_{\pm 1.6}$ | $70.8_{\pm 0.8}$ |
| DFR [33, 28] | ✓ | ✓ | $91.1_{\pm 0.8}$ | $89.4_{\pm 0.9}$ | $78.8_{\pm 0.5}$ | $72.6_{\pm 0.3}$ |
| Group DRO-ES [58, 28] | ✓ | ✓ | $90.7_{\pm 0.6}$ | $90.6_{\pm 1.6}$ | $80.4$ | $73.5$ |

**Spurious correlations.** The performance of empirical risk minimization (ERM) in the presence of spurious correlations has been extensively studied [18]. In vision, ERM models are widely known to rely on spurious attributes like background [58, 75], texture [17], and secondary objects [56, 61, 62] to perform classification. In language, ERM models often utilize syntactic or statistical heuristics as a substitute for semantic understanding [20, 50, 46]. This behavior can lead to bias against demographic minorities [25, 6, 67, 23, 8] or failure in high-consequence applications [42, 9, 79, 51].

**Robustness and group annotations.** If group annotations are available in the training dataset, *group distributionally robust optimization* (DRO) [58] can improve robustness by minimizing the worst-group loss, while other techniques learn invariant or diverse features [1, 19, 80, 76]. Methods which use only partial group annotations include *deep feature reweighting* (DFR) [33, 28], which retrains the last layer on a group-balanced held-out set, and *spread spurious attribute* [48], which performs DRO with group pseudo-labels. Recently, more lightweight methods that only adjust the model predictions using a group annotated held-out set have also been proposed [70]. However, since the groups are often unknown ahead of time or difficult to annotate in practice, there has been significant interest in methods which do not utilize group annotations except for model selection. The bulk of these techniques utilize auxiliary models to pseudo-label the minority group or spurious feature [63, 47, 77, 11, 31, 66, 32, 64, 81]; notably, *just train twice* (JTT) [41] upweights samples misclassified by an early-stopped model. Other techniques reweight or subsample the classes [27] or train with robust losses and regularization [54, 78].

Unlike DFR, our methods utilize no held-out group annotations, and unlike JTT, we do not have to "train twice", only finetune the last layer. We also show that SELF performs best using *disagreements* between the ERM and early-stopped models instead of *misclassifications* as in JTT. Moreover, while JTT assumes the early-stopped model has low worst-group accuracy which improves during training, SELF performs well even when the early-stopped model has high worst-group accuracy which decreases during training – substantially improving performance on datasets such as CivilComments. The concurrent work of Qiu *et al.* [55] proposed a similar method to our Section 4.2; while they use a tunable loss to upweight misclassified samples, our method shows that similar results can be achieved with no hyperparameter tuning (and therefore no group annotations for the validation set). Finally, while our results partially corroborate the findings of Idrissi *et al.* [27] that class balancing during ERM is effective for group robustness, we observe that class-balanced last-layer retraining renders ERM class balancing optional. We compare our results with previous methods in Table 1.

**Generalization via disagreement.** Disagreement-based active learning for improving in-distribution generalization has been well-studied since before the deep learning era [10, 4, 21]. More recent research has utilized disagreements between SGD runs to predict in-distribution gen-

Table 2: **Dataset composition.** We study four well-established benchmarks for group robustness across vision and language tasks. The class probabilities change dramatically when conditioned on the spurious feature. Note that Waterbirds is the only dataset that has a distribution shift and MultiNLI is the only dataset which is class-balanced *a priori*. Probabilities may not sum to 1 due to rounding.

| Dataset | Group $g$ | | Training distribution $\hat{p}$ | | | Data quantity | | |
|---|---|---|---|---|---|---|---|---|
| | Class $y$ | Spurious $s$ | $\hat{p}(y)$ | $\hat{p}(g)$ | $\hat{p}(y\|s)$ | Train | Val | Test |
| Waterbirds | landbird | land | .768 | .730 | .984 | 3498 | 467 | 2225 |
| | landbird | water | | .038 | .148 | 184 | 466 | 2225 |
| | waterbird | land | .232 | .012 | .016 | 56 | 133 | 642 |
| | waterbird | water | | .220 | .852 | 1057 | 133 | 642 |
| CelebA | non-blond | female | .851 | .440 | .758 | 71629 | 8535 | 9767 |
| | non-blond | male | | .411 | .980 | 66874 | 8276 | 7535 |
| | blond | female | .149 | .141 | .242 | 22880 | 2874 | 2480 |
| | blond | male | | .009 | .020 | 1387 | 182 | 180 |
| CivilComments | neutral | no identity | .887 | .551 | .921 | 148186 | 25159 | 74780 |
| | neutral | identity | | .336 | .836 | 90337 | 14966 | 43778 |
| | toxic | no identity | .113 | .047 | .079 | 12731 | 2111 | 6455 |
| | toxic | identity | | .066 | .164 | 17784 | 2944 | 8769 |
| MultiNLI | contradiction | no negation | .333 | .279 | .300 | 57498 | 22814 | 34597 |
| | contradiction | negation | | .054 | .761 | 11158 | 4634 | 6655 |
| | entailment | no negation | .334 | .327 | .352 | 67376 | 26949 | 40496 |
| | entailment | negation | | .007 | .104 | 1521 | 613 | 886 |
| | neither | no negation | .333 | .323 | .348 | 66630 | 26655 | 39930 |
| | neither | negation | | .010 | .136 | 1992 | 797 | 1148 |

eralization [30], as well as between model classes (*e.g.,* CNNs and Transformers) to predict out-of-distribution generalization [3]. Two works that are concurrent to ours, *diversity by disagreement* [52] and *diversify and disambiguate* [40], are methods for generalization under distribution shift which maximize the disagreement between multiple predictors to learn a diverse ensemble. Compared to our work, these methods are optimized for training datasets which exhibit a *complete correlation* (*i.e.,* contain no minority group data) and they underperform in the less extreme spurious correlation setting we study.

## 3   Preliminaries

**Setting.** We consider classification tasks with input domain $\mathcal{X}$ and target classes $\mathcal{Y}$. We assume $\mathcal{S}$ is a set of *spurious features* such that each sample $x \in \mathcal{X}$ is associated with exactly one feature $s \in \mathcal{S}$. In conjunction with the target class, the spurious features partition the dataset into groups $\mathcal{G} = \mathcal{Y} \times \mathcal{S}$. While the groups may be heavily imbalanced in the training distribution, we desire a model which is invariant to the spurious feature and thus has roughly uniform performance over $\mathcal{G}$. Therefore, we evaluate *worst-group accuracy* (WGA), *i.e.,* the minimum accuracy among all groups [58].

We will often refer to datasets and models as *group-balanced* or *class-balanced*, meaning that in expectation, the dataset is composed of an equal number of samples from groups in $\mathcal{G}$ or classes in $\mathcal{Y}$, respectively. This balance can be achieved by training on a subset with equal data from each group/class, or sampling from the data so that each minibatch is balanced in expectation [27]. To make the latter more concrete, for group balancing we first sample $s \sim \text{Unif}(\mathcal{S})$, then sample $x \sim \hat{p}(\cdot|s)$ where $\hat{p}$ is the training distribution; class balancing is the same with $\mathcal{Y}$ instead of $\mathcal{S}$. We use the minibatch sampling approach for both class-balanced ERM and class-balanced last-layer retraining, and we provide a comparison with the subset method in Appendix A.

**Deep feature reweighting.**   The recently proposed *deep feature reweighting* (DFR) [33, 28] method achieves state-of-the-art WGA by performing ERM on the training dataset, then retraining the last layer of the neural network on a group-balanced held-out dataset, called the *reweighting dataset*. In the original implementation, half the validation set is used to construct the reweighting dataset: all data from the smallest group is included and the other groups are randomly downsampled to that size.

Then, the feature embeddings (*i.e.,* the outputs of the penultimate layer) of the reweighting dataset are pre-computed and used to train a logistic regression model with explicit $\ell_1$-regularization. The results are averaged over 10 randomly subsampled reweighting datasets, and a hyperparameter search is performed over $\ell_1$ regularization strength on the other half of the validation set.

To emphasize practicality and efficiency, our implementation of DFR has some differences from the original. *(i)* Instead of logistic regression, we train the last layer on the reweighting dataset via minibatch optimization using SGD and AdamW [43] for the vision and language tasks, respectively. This fits well into standard training pipelines and avoids pre-computing the feature embeddings and writing them to disk, which can be slow and memory-intensive. *(ii)* To reduce the number of hyperparameters, we use a fixed-value $\ell_2$ regularization instead of searching over $\ell_1$ regularization strength. We observed similar performance for $\ell_1$ and $\ell_2$ regularization, which we believe is because $\ell_1$-regularized gradients do not induce sparsity [39]. *(iii)* We sample uniformly at random from the groups in the held-out dataset (to get group balanced minibatches) instead of averaging over group-balanced subsets of the data [27]. We compare the implementations in detail in Appendix A.

**Datasets and models.** We study four datasets which are well-established as benchmarks for group robustness across vision and language tasks, detailed in Table 2 and summarized below.

- *Waterbirds* [71, 69, 58] is an image classification dataset where the task is to predict whether a bird is a landbird or a waterbird. The spurious feature is the image background: more landbirds are present on land backgrounds than waterbirds, and vice versa.[2]

- *CelebA* [42, 58] is an image classification dataset where the task is to predict whether a person is blond or not. The spurious feature is gender, with $16\times$ more blond women than blond men in the training set.

- *CivilComments* [7, 35] is a text classification dataset where the task is to predict whether a comment is toxic or not. The spurious feature is the presence of one of the following categories: male, female, LGBT, black, white, Christian, Muslim, or other religion.[3] More toxic comments contain one of these categories than non-toxic comments, and vice versa.

- *MultiNLI* [73, 58] is a text classification dataset where the task is to predict whether a pair of sentences is a contradiction, entailment, or neither. The spurious feature is a negation in the second sentence – more contradictions have this property than entailments or neutral pairs.

Importantly, Waterbirds is the only dataset that has a distribution shift – its validation and test datasets are group-balanced conditioned on the classes, although still class-imbalanced.[4] As a result, methods which use the validation set for training can improve performance without explicit group balancing.

We utilize a ResNet-50 [24] pretrained on ImageNet-1K [57] for Waterbirds and CelebA, and a BERT [14] model pretrained on Book Corpus [83] and English Wikipedia for CivilComments and MultiNLI. Following previous work, we use half the validation set for feature reweighting [33, 28] and half for model selection with group annotations [58, 41, 33, 48, 28]. We run each experiment on three random seeds and do not utilize explicit early stopping except as part of SELF (see Section 5). See Appendix D for further details on the training procedure.

## 4 Class-balanced last-layer retraining

In this section, we investigate the necessity of a group-balanced reweighting dataset for DFR and show that *class-balanced last-layer retraining* is a simple but strong baseline for group robustness without group annotations. To enable a fair comparison with our implementation of DFR (see Section 3), class-balanced last-layer retraining follows the same training procedure, except the reweighting dataset is constructed by sampling uniformly over the classes $\mathcal{Y}$ instead of the groups $\mathcal{G}$.

---

[2]We note that the Waterbirds dataset is known to contain incorrect labels [66]. We report results on the original, un-corrected version for a fair comparison with previous work.

[3]This version of CivilComments has four groups, used in this work and by [58, 28, 33]. There is another version where the identity categories are not collapsed into one spurious feature; this version is used by [41, 81], so we do not report their CivilComments accuracies in Table 1. Both versions use the WILDS split [35].

[4]The Waterbirds validation and test datasets still contain more landbirds than waterbirds, but each class has equal quantities of land and water backgrounds.

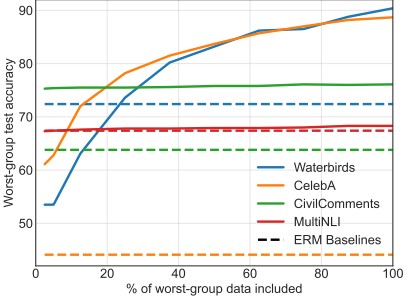

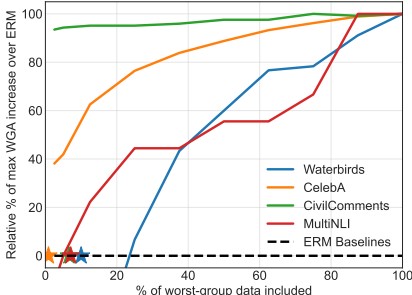

(a) Worst-group test accuracy

(b) Relative % of max WGA increase over ERM

Figure 1: **How much worst-group data does last-layer retraining really need?** We perform an ablation on the percentage of worst-group data used for class-balanced last-layer retraining, while keeping the total data constant. The results show that last-layer retraining can substantially improve worst-group accuracy *even when the reweighting dataset has only a small proportion of worst group data*, and that class balancing can be a major factor in its performance. The underperformance on Waterbirds below 20% is because there is too little worst-group data to observe consistent model behavior (less than ten samples). The stars ★ denote the baseline percentage of worst-group data in the training dataset. We plot the mean over three independent runs. See Tables 12 and 13 for details.

Table 3: **Last-layer retraining on the held-out dataset.** While unbalanced (UB) last-layer retraining decreases performance, class-balanced (CB) last-layer retraining nearly matches DFR on Waterbirds and CivilComments. However, it still trails DFR on CelebA and MultiNLI; we improve these results with the SELF method described in Section 5. CB ERM is trained on the combined training and held-out datasets using class-balanced minibatches. We list the mean and standard deviation over three independent runs.

| Method | Group annotations | Worst-group test accuracy | | | |
|---|---|---|---|---|---|
| | | Waterbirds | CelebA | CivilComments | MultiNLI |
| Class-balanced ERM | ✗ | $81.9_{\pm 3.4}$ | $67.2_{\pm 5.6}$ | $61.4_{\pm 0.7}$ | $\mathbf{69.2_{\pm 1.6}}$ |
| UB last-layer retraining | ✗ | $88.0_{\pm 0.8}$ | $41.9_{\pm 1.4}$ | $57.6_{\pm 4.2}$ | $64.6_{\pm 1.0}$ |
| CB last-layer retraining | ✗ | $\mathbf{92.6_{\pm 0.8}}$ | $\mathbf{73.7_{\pm 2.8}}$ | $\mathbf{80.4_{\pm 0.8}}$ | $64.7_{\pm 1.1}$ |
| DFR (our impl.) | ✓ | $92.4_{\pm 0.9}$ | $87.0_{\pm 1.1}$ | $81.8_{\pm 1.6}$ | $70.8_{\pm 0.8}$ |
| DFR [33, 28] | ✓ | $91.1_{\pm 0.8}$ | $89.4_{\pm 0.9}$ | $78.8_{\pm 0.5}$ | $72.6_{\pm 0.3}$ |

## 4.1 An ablation on the proportion of worst-group data

In this section, we perform an ablation on the percentage of worst-group data in the reweighting dataset and show that class-balanced last-layer retraining can substantially improve WGA *even when the reweighting dataset has only a small proportion of worst group data.* For our ablation, we choose the worst groups based on the performance of ERM, and if two groups have similarly poor performance, we vary both. The worst groups for Waterbirds are landbirds on water backgrounds and waterbirds on land backgrounds; for CelebA blond men; for CivilComments toxic comments containing identity categories; and for MultiNLI entailments and neutral pairs containing negations.

For these experiments, we begin with the DFR reweighting dataset, *i.e.,* a random group-balanced subset of the held-out dataset, where the size of each group is the minimum of the worst-group size and half the size of any other group.[5] We define this dataset to include 100% of the worst-group data. We then reduce the percentage of worst-group data, while correspondingly increasing the percentage of data from the same class without the spurious feature. For example, the DFR reweighting dataset on CelebA has 92 points from each group, so reducing the worst group to 25% results in 92 non-blond females, 92 non-blond males, 161 blond females, and 23 blond males. The total data is kept constant.

---

[5]This is necessary because we keep the total data constant. By reducing the quantity of worst-group data to zero, we will at most double the size of any other group.

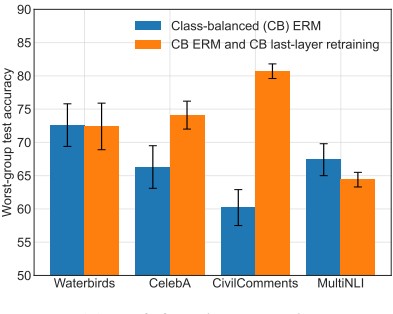

(a) Training dataset only

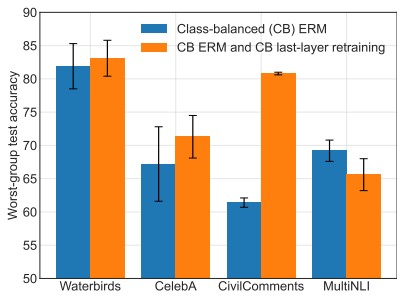

(b) Combined training and held-out datasets

Figure 2: **A "free lunch" in group robustness.** We compare class-balanced (CB) ERM on the entire dataset to splitting the dataset and performing CB ERM on the first (95%) split and CB last-layer retraining on the second (5%) split. This technique improves worst-group accuracy on Waterbirds, CelebA, and CivilComments by up to 17% while using *no additional data or annotations* for training beyond ERM. We believe it underperforms on MultiNLI because there is not enough data in the first split, *i.e.,* ERM performance can be improved by collecting more data. We plot the mean and standard deviation over three independent runs. See Table 14 for detailed results.

The results of this ablation are displayed in Figure 1. While a smooth increase in WGA with increasing worst-group data is expected, the extent of the early increase is surprising: over the four datasets, an average of 67% of the increase in WGA over class-unbalanced ERM is obtained by the first 25% of worst-group data. In particular, CelebA and CivilComments – the most class-imbalanced datasets we study – experience significant improvement even at low percentages of worst-group data. This phenomenon suggests that, while group balancing is still important for best results, *class balancing is a major factor in the performance of DFR* on these two datasets.

Furthermore, class-balanced last-layer retraining can improve worst-group accuracy even when the held-out dataset has similar group imbalance as the training dataset (*i.e.,* at the stars ★ in Figure 1). Based on this observation, we propose that class-balanced last-layer retraining *on the entire held-out dataset* can be a simple but strong baseline for group robustness without group annotations. Table 3 details the results; on average over the four datasets, class-balanced last-layer retraining achieves 94% of DFR performance, compared to 76% without class balancing. However, it still trails DFR by a significant amount on CelebA and MultiNLI – the most group-imbalanced datasets we study. We improve these results with our selective last-layer finetuning (SELF) method described in Section 5.

Moreover, our experiments in Figure 4 (deferred to Appendix A) indicate that class-balanced last layer retraining has similar performance regardless of whether it is initialized with class-unbalanced or class-balanced ERM features. Contrasting with Idrissi *et al.* [27], our results suggest that class balancing in the ERM stage is optional. This result has practical relevance for expensive models pre-trained without class balancing (*e.g.,* large language models), as the benefits of class balancing on downstream tasks can be reaped by simply retraining the last layer instead of training a new model.

## 4.2 A "free lunch" in group robustness

Motivated by the promising results of Section 4.1, where we performed class-balanced last-layer retraining on a fixed held-out set, we now ask *how can we best utilize a realistic training dataset?* In particular, practical applications often work with a predetermined data, annotation, and compute budget. Within this budget, and with no explicit held-out dataset, would one achieve better group robustness by using the entire dataset for ERM or by holding out a subset for last-layer retraining?

We investigate this question on our four benchmark datasets by randomly splitting the initial dataset into two, then performing class-balanced ERM training on the first split (95% of the data) and class-balanced last-layer retraining on the second split (5% of the data). Figure 2a illustrates the results of our experiments on the training dataset and Figure 2b on the combined training and held-out datasets. Since the quantity of data is higher in the case of combining the training and held-out datasets, we expect all numbers to be higher in Figure 2b compared to Figure 2a (especially on

Table 4: **Comparison of selective last-layer finetuning methods.** SELF nearly matches DFR and improves WGA over class-balanced (CB) last-layer retraining by up to 12%. Early-stop (ES) disagreement SELF performs the best overall, and disagreement methods perform especially well on CivilComments. Dropout and ES disagreement request few as 20 class annotations from the held-out dataset. All SELF methods use a small set of group annotations for model selection. We list the mean and standard deviation over three independent runs.

| Method | Held-out annotations | | Worst-group test accuracy | | | |
|---|---|---|---|---|---|---|
| | Group | Class | Waterbirds | CelebA | CivilComments | MultiNLI |
| Class-balanced ERM | ✗ | ✓ | $81.9_{\pm 3.4}$ | $67.2_{\pm 5.6}$ | $61.4_{\pm 0.7}$ | $69.2_{\pm 1.6}$ |
| CB last-layer retraining | ✗ | ✓ | $92.6_{\pm 0.8}$ | $73.7_{\pm 2.8}$ | $80.4_{\pm 0.8}$ | $64.7_{\pm 1.1}$ |
| Random SELF | ✗ | ✗ | $91.1_{\pm 1.9}$ | $80.2_{\pm 6.4}$ | $\mathbf{80.5_{\pm 1.6}}$ | $65.0_{\pm 4.0}$ |
| Misclassification SELF | ✗ | ✓ | $92.6_{\pm 0.8}$ | $83.0_{\pm 6.1}$ | $62.7_{\pm 4.6}$ | $72.2_{\pm 2.2}$ |
| ES misclassification SELF | ✗ | ✓ | $92.2_{\pm 0.7}$ | $80.4_{\pm 3.9}$ | $65.8_{\pm 7.6}$ | $\mathbf{73.3_{\pm 1.2}}$ |
| Dropout disagreement SELF | ✗ | ✗ | $92.3_{\pm 0.5}$ | $\mathbf{85.7_{\pm 1.6}}$ | $69.9_{\pm 5.2}$ | $68.7_{\pm 3.4}$ |
| ES disagreement SELF | ✗ | ✗ | $\mathbf{93.0_{\pm 0.3}}$ | $83.9_{\pm 0.9}$ | $79.1_{\pm 2.1}$ | $70.7_{\pm 2.5}$ |
| DFR (our impl.) | ✓ | ✓ | $92.4_{\pm 0.9}$ | $87.0_{\pm 1.1}$ | $81.8_{\pm 1.6}$ | $70.8_{\pm 0.8}$ |
| DFR [33, 28] | ✓ | ✓ | $91.1_{\pm 0.8}$ | $89.4_{\pm 0.9}$ | $78.8_{\pm 0.5}$ | $72.6_{\pm 0.3}$ |

Waterbirds, which has a more group-balanced validation set). We perform no hyperparameter tuning for these experiments, and therefore we do not utilize group annotations *even for model selection*.

Figure 2 indicates that splitting the dataset and performing last-layer retraining substantially improves worst-group accuracy on Waterbirds, CelebA, and CivilComments. It decreases performance on MultiNLI, which is the only dataset where adding held-out data from the same distribution significantly increases ERM worst-group accuracy compared to DFR. Specifically, class-balanced ERM achieves $67.4 \pm 2.4\%$ WGA on the training dataset and $69.2 \pm 1.6\%$ on the combined training and held-out datasets, while our DFR implementation achieves $70.8 \pm 0.8\%$. Therefore, we hypothesize that last-layer retraining on the second split can improve group robustness *only if there is enough data for ERM to perform near-optimally* on the first split, *i.e.,* if the performance of ERM on the first split is limited by dataset bias rather than sample variance.

Based on this hypothesis, our answer to the posed question is: if ERM performance is stable when holding out $5\%$ of data, perform last-layer retraining on the held-out dataset instead of ERM on the initial dataset. We call this technique a "free lunch" because it improves worst-group accuracy with *no additional data or annotations* beyond ERM (including for model selection). In particular, we utilize less data for ERM training and less compute due to the efficient nature of last-layer retraining. Therefore, we believe this method is especially relevant to practitioners, and it can be easily implemented with little change to data processing or model training workflows.

A critical remaining question is *why* last-layer retraining improves group robustness; since the training and held-out datasets have equally drastic group imbalance, it is counterintuitive that reducing the quantity of data used for ERM and performing last-layer retraining would increase worst-group accuracy. In some cases, as seen on MultiNLI in Figure 2, training an ERM model with less data can be detrimental – but on the other three datasets, last-layer retraining substantially improves over ERM. We leave it to future empirical and theoretical work to better understand this phenomenon.

## 5 Selective last-layer finetuning

While class-balanced last-layer retraining can improve worst-group accuracy without group annotations, its performance is still inferior to DFR, particularly on the highly group-imbalanced CelebA and MultiNLI datasets. In these cases, training on the entire held-out set can be detrimental; instead, we show that constructing the reweighting dataset by detecting and upsampling worst-group data is more effective. We propose *selective last-layer finetuning* (SELF), which uses an auxiliary model to select a small, more group-balanced reweighting dataset, then *finetunes* the ERM last layer *instead of retraining entirely* to avoid overfitting on this smaller dataset.

SELF does not use group annotations for training and can be implemented even when class annotations are unavailable in the held-out dataset, making it applicable in settings not captured by current techniques (*e.g.,* adaptation to an unlabeled target domain [45]). If class annotations *are not* available, we select points which are disagreed upon by ERM and a regularized model, then request their labels. If class annotations *are* available, we can still use disagreement, or alternatively, we select points which are misclassified by the ERM or early-stopped model. Our experiments indicate that *disagreement SELF between the ERM and early-stopped models* performs the best overall, increasing the worst-group accuracy of class-balanced last-layer retraining to near-DFR levels while requesting less than 3% of the held-out class annotations. Our results are detailed in Table 4.

We formalize SELF as follows. Let $f, g : \mathcal{X} \to \mathbb{R}^{|\mathcal{Y}|}$ be models with logit outputs, $X \subseteq \mathcal{X}$ be a held-out dataset, $c : \mathbb{R}^{2|\mathcal{Y}|} \to \mathbb{R}$ be a cost function, and $n$ be an integer. SELF constructs the reweighting set $D$ by greedily selecting the $n$ points in $X$ whose outputs incur the highest cost, *i.e.,*

$$D = \operatorname*{argmax}_{S \subseteq X, |S|=n} \sum_{x \in S} c(f(x), g(x)). \tag{1}$$

Then, SELF requests class annotations and performs class-balanced *finetuning* (*i.e.,* starting from the ERM weights) of the last layer of the ERM model on $D$, using the same implementation as last-layer retraining (see Section 4). We study the following four variants of SELF:

- *Misclassification*: $f$ is an ERM model and $g$ is the labeling function, *i.e.,* , for a datapoint $x$ with label $y$, the output $g(x)$ is the one-hot encoded vector for $y$. The cost function $c$ is cross entropy loss.
- *Early-stop misclassification*:[6] $f$ is an early-stopped ERM model and $g$ is the labeling function. The cost function $c$ is cross entropy loss.
- *Dropout disagreement*: $f$ is an ERM model and $g$ is the same model where inference is run with nodes randomly dropped out in the last layer [65]. The cost function $c$ is KL divergence with softmax.
- *Early-stop disagreement*: $f$ is an ERM model and $g$ is an early-stopped ERM model. The cost function $c$ is KL divergence with softmax.

We also experiment with a pure random baseline, referred as Random SELF in Table 4, where $D$ consists of $n$ random points from $X$. While Random SELF suffers high variance as expected, perhaps surprisingly its performance is still competitive on average. The reweighting dataset size $n$ is a hyperparameter, and for disagreement SELF it corresponds to the quantity of class annotations requested. Please see Appendices B and D for hyperparameter details and ablation studies.

## 5.1  Analysis of SELF performance

To quantify the extent of SELF's upsampling, we plot the percentage of the reweighting dataset consisting of worst-group data in Figure 3. In particular, we present to the best of our knowledge the first evidence that model disagreement effectively upsamples worst-group data, an observation which may be of independent interest. With that said, Figure 3 does not formally establish that better group balance has a strong correlation with WGA – rather, it suggests an *additional subtlety in the DFR hypothesis*, since group balance alone does not fully explain the performance of last-layer retraining. In addition to the balance of the reweighting dataset, it is likely that characteristics of the *specific data selected* also contribute to SELF results. Additionally, the competitive performance of pure random SELF in Table 4 further questions the importance of group balancing in DFR.

The importance of which data are selected could explain why disagreement SELF often outperforms misclassification SELF despite having access to less information. While misclassification selects the *most difficult* or *noisiest* points, disagreement selects the *most uncertain* points. For example, dropout models approximate a theoretically justified uncertainty metric [16] which is likely to be higher on worst-group data, and early-stopped models tend to fit simple patterns first [2, 41] including the majority group. Training on the most uncertain data is a key tenet of active learning [13, 59, 12], which rationalizes the performance of disagreement SELF and provides motivation for further investigation of *why* last-layer retraining improves group robustness.

---

[6]Using early-stop misclassification to upsample worst-group data is the premise of JTT [41].

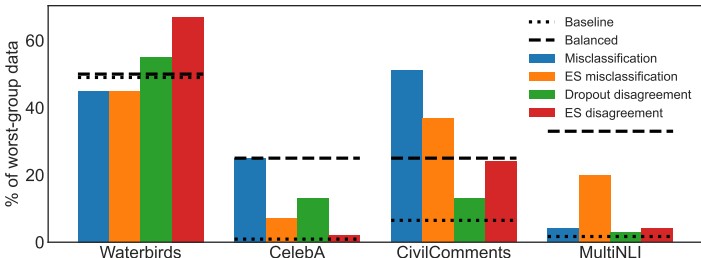

Figure 3: **SELF upsamples the worst group.** We plot the percentage of the reweighting dataset consisting of worst-group data for each SELF method. "Baseline" represents the percentage of worst-group data in the held-out dataset, while "Balanced" is the percentage of worst-group data necessary to achieve group balance. This is to the best of our knowledge the first empirical evidence that model disagreement is an effective method for upsampling worst-group data. The worst groups in each dataset are listed in Section 4.1. We plot the mean for the best dataset size $n$ over three independent runs. See Table 15 for detailed results.

A remaining question is why the performance of disagreement SELF is so strong on CivilComments compared to misclassification-based methods such as JTT [41] and ES misclassification SELF. We show in Figure 6 (deferred to Appendix B) that, contrary to the assumptions made by JTT and other early-stop misclassification methods, the worst-group accuracy *decreases* with training on CivilComments. This highlights a potential advantage of disagreement methods over misclassification methods, as disagreement is justified *regardless of whether the regularized or ERM model has a greater dependence on spurious features*. Moreover, while misclassification methods do indeed upsample the minority group in Figure 3, we show in Table 10 (deferred to Appendix B) that the held-out set training accuracy of misclassification SELF tends to be very low – down to $0\%$ on MultiNLI – evidence that misclassifications are too difficult to learn without modifying the features.

**Theoretical proof-of-concept.** The observations of Section 5.1 raise the question of whether disagreement SELF can ever be shown to *provably upsample minority group points*. We provide a simple proof-of-concept for SELF by considering the last layer to be a linear model (which could be under- or overparameterized) with core, spurious and junk features. We show that the disagreement between the regularized model and the ERM model (measured by total variation distance between the predicted distributions) is provably higher on minority examples than majority examples, *regardless of model dependence on the spurious feature*. In particular, this shows provable benefits of disagreement SELF even in situations where the early-stopped model has higher worst-group accuracy than the convergent model, which may not be captured by related methods in the literature [41]. Our detailed setup, assumptions, and main theoretical result (Theorem 1) are stated in detail in Appendix C.

## 6 Conclusion

In this paper, we presented a comprehensive examination of the performance of last-layer retraining in the absence of group and class annotations. We showed that class-balanced last-layer retraining is a simple but strong baseline for group robustness, and that holding out a subset of the training data to retrain the last layer can substantially improve worst-group accuracy. We then proposed selective last-layer finetuning (SELF), whose early-stop disagreement version improves performance to near-DFR levels with no group annotations and less than $3\%$ of the held-out class annotations.

Our work has generated several open questions which could prove fruitful for further research. First, why does last-layer retraining on a held-out split of the training dataset improve group robustness (see Section 4.2)? Second, to what extent does the precise data selected matter for reweighting, *e.g.,* which of disagreement SELF and misclassification SELF would perform better when hyperparameters are set to equalize group balance? Third, what is the optimal strategy in general for obtaining the regularized model in disagreement SELF (dropout, early-stopping, or a different technique) and why?

**Acknowledgments.** We thank Google Cloud for the gift of compute credits, Jacob Abernethy and Eva Dyer for the additional compute assistance, and anonymous reviewers for their helpful feedback. We also thank Harikrishna Narasimhan for reading an earlier version of the manuscript and providing thoughtful comments and Seokhyeon Jeong for catching an error in an earlier version of the code. T.L. acknowledges support from the DoD NDSEG Fellowship. V.M. acknowledges support from the NSF (through CAREER award CCF-2239151 and award IIS-2212182), an Adobe Data Science Research Award, an Amazon Research Award, and a Google Research Collabs Award.

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

## Appendix Outline

The appendix is organized as follows. In Appendix A we include additional experiments for Sections 3 and 4. In Appendix B we include additional experiments for Section 5. In Appendix C we include the statement and proof of our theoretical proof-of-concept (Theorem 1). In Appendix D we describe in detail our training procedure and hyperparameters. Appendix E include tables to accompany the figures in the main paper. Finally, Appendix F discusses broader impacts, limitations, and compute.

## A  Additional experiments for Sections 3 and 4

In this section we include the detailed results of additional experiments for Sections 3 and 4.

Table 5: **Empirical risk minimization.** First, we compare class-unbalanced (CU) and class-balanced (CB) ERM on the training dataset vs. the combined training and held-out datasets (*i.e.,* the training dataset plus half the validation dataset). We list the mean and standard deviation over three independent runs. Note that the Waterbirds validation dataset has a different distribution than the training dataset (and, in particular, is group-balanced), and that MultiNLI is class-balanced *a priori*.

| Method | Held-out dataset included | Worst-group test accuracy | | | |
|--------|---------------------------|------------|--------|--------------|----------|
| | | Waterbirds | CelebA | CivilComments | MultiNLI |
| CU ERM | ✗ | $72.4_{\pm 1.0}$ | $44.1_{\pm 0.9}$ | $63.8_{\pm 6.2}$ | $67.4_{\pm 2.4}$ |
| CB ERM | ✗ | $72.6_{\pm 3.2}$ | $66.3_{\pm 3.2}$ | $60.2_{\pm 2.7}$ | $67.4_{\pm 2.4}$ |
| CU ERM | ✓ | $81.6_{\pm 1.5}$ | $44.5_{\pm 3.4}$ | $59.1_{\pm 2.2}$ | $69.1_{\pm 1.3}$ |
| CB ERM | ✓ | $81.9_{\pm 3.4}$ | $67.2_{\pm 5.6}$ | $61.4_{\pm 0.7}$ | $69.2_{\pm 1.6}$ |

Table 6: **Balancing methodology comparison.** Next, we compare last-layer retraining on the held-out set with different balancing methodologies, each initialized with class-balanced ERM features. In "sampling", we sample from the held-out dataset at a non-uniform rate so that each minibatch is class- or group-balanced in expectation, while in "subset", we train on a random class- or group-balanced subset of the held-out dataset. Specifically, for group-balanced sampling we first sample $s \sim \text{Unif}(\mathcal{S})$, then sample $x \sim \hat{p}(\cdot|s)$ where $\hat{p}$ is the training distribution; class-balanced sampling is the same with $\mathcal{Y}$ instead of $\mathcal{S}$. For subset balancing, we keep all data from the smallest group/class and downsample the others uniformly at random to that size. We list the mean and standard deviation over three independent runs.

| Last-layer retraining method | Worst-group test accuracy | | | |
|------------------------------|------------|--------|--------------|----------|
| | Waterbirds | CelebA | CivilComments | MultiNLI |
| Class-unbalanced | $88.0_{\pm 0.8}$ | $41.9_{\pm 1.4}$ | $57.6_{\pm 4.2}$ | $64.6_{\pm 1.0}$ |
| Class-balanced sampling | $92.6_{\pm 0.8}$ | $73.7_{\pm 2.8}$ | $80.4_{\pm 0.8}$ | $64.7_{\pm 1.1}$ |
| Class-balanced subset | $92.1_{\pm 0.9}$ | $74.6_{\pm 2.0}$ | $80.2_{\pm 1.4}$ | $64.5_{\pm 1.3}$ |
| Group-balanced sampling | $92.4_{\pm 0.9}$ | $87.0_{\pm 1.1}$ | $81.8_{\pm 1.6}$ | $70.8_{\pm 0.8}$ |
| Group-balanced subset | $91.6_{\pm 1.9}$ | $88.1_{\pm 1.0}$ | $79.2_{\pm 1.8}$ | $68.3_{\pm 1.8}$ |
| DFR [33, 28] | $91.1_{\pm 0.8}$ | $89.4_{\pm 0.9}$ | $78.8_{\pm 0.5}$ | $72.6_{\pm 0.3}$ |

Table 7: **Necessity of the held-out dataset.** Next, we investigate whether holding out a subset of the training dataset for class-balanced (CB) last-layer retraining is essential, or if retraining on the entire (previously seen) dataset is also effective. For retraining on the training dataset, we use the same class-balanced last-layer retraining procedure as the held-out dataset, but we train for 20 epochs for the vision tasks and 2 epochs for the language tasks. For retraining on the held-out dataset, we report the best over four splits (*i.e.,* the same numbers as Figure 2). Our results suggest that holding out data is necessary to achieve maximal worst-group accuracy, though last-layer retraining on the training dataset interestingly prevents the performance decrease on MultiNLI. We list the mean and standard deviation over three independent runs.

| Method | Worst-group test accuracy | | | |
| --- | --- | --- | --- | --- |
| | Waterbirds | CelebA | CivilComments | MultiNLI |
| CB ERM | $72.6_{\pm3.2}$ | $66.3_{\pm3.2}$ | $60.2_{\pm2.7}$ | $67.4_{\pm2.4}$ |
| CB last-layer retraining on training dataset | $71.0_{\pm2.5}$ | $66.9_{\pm1.4}$ | $61.9_{\pm0.8}$ | $67.0_{\pm1.5}$ |
| CB last-layer retraining on held-out dataset | $77.4_{\pm0.3}$ | $73.0_{\pm2.3}$ | $77.9_{\pm1.5}$ | $63.0_{\pm1.5}$ |

Table 8: **Average accuracy performance.** We detail the average test accuracy of our methods on the 4 benchmark datasets. Both of our methods have similar average accuracy to DFR, which experiences a slight accuracy/robustness tradeoff compared to ERM (as is typical in the robustness literature). We list the mean and standard deviation over three independent seeds.

| Method | Group Anns | Average test accuracy | | | |
| --- | --- | --- | --- | --- | --- |
| | | Waterbirds | CelebA | CivilComments | MultiNLI |
| ERM | ✗ | $90.2_{\pm0.7}$ | $94.4_{\pm0.2}$ | $92.0_{\pm0.2}$ | $81.8_{\pm0.2}$ |
| CB last-layer retraining | ✗ | $94.8_{\pm0.3}$ | $93.6_{\pm0.2}$ | $87.1_{\pm0.0}$ | $82.0_{\pm0.2}$ |
| ES disagreement SELF | ✗ | $94.0_{\pm1.7}$ | $91.7_{\pm0.4}$ | $87.7_{\pm0.6}$ | $81.2_{\pm0.7}$ |
| DFR (our impl.) | ✓ | $94.9_{\pm0.3}$ | $92.6_{\pm0.5}$ | $87.5_{\pm0.2}$ | $81.7_{\pm0.2}$ |

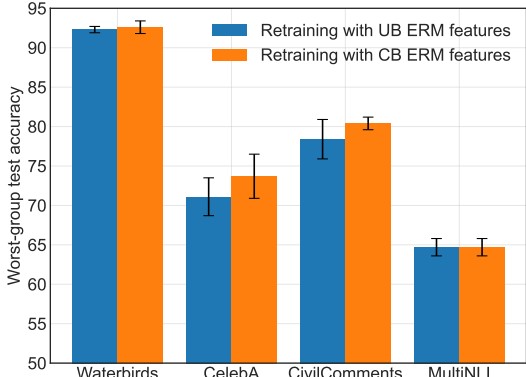

Figure 4: **Class-balanced last-layer retraining renders class-balanced ERM optional.** We compare class-balanced (CB) last-layer retraining on the held-out dataset initialized with unbalanced (UB) or CB ERM features. Our results show that CB ERM is unnecessary as long as the last layer is retrained with class balancing. We plot the mean and standard deviation over three independent runs.

# B   Additional experiments for Section 5

In this section we include the detailed results of additional experiments for Section 5.

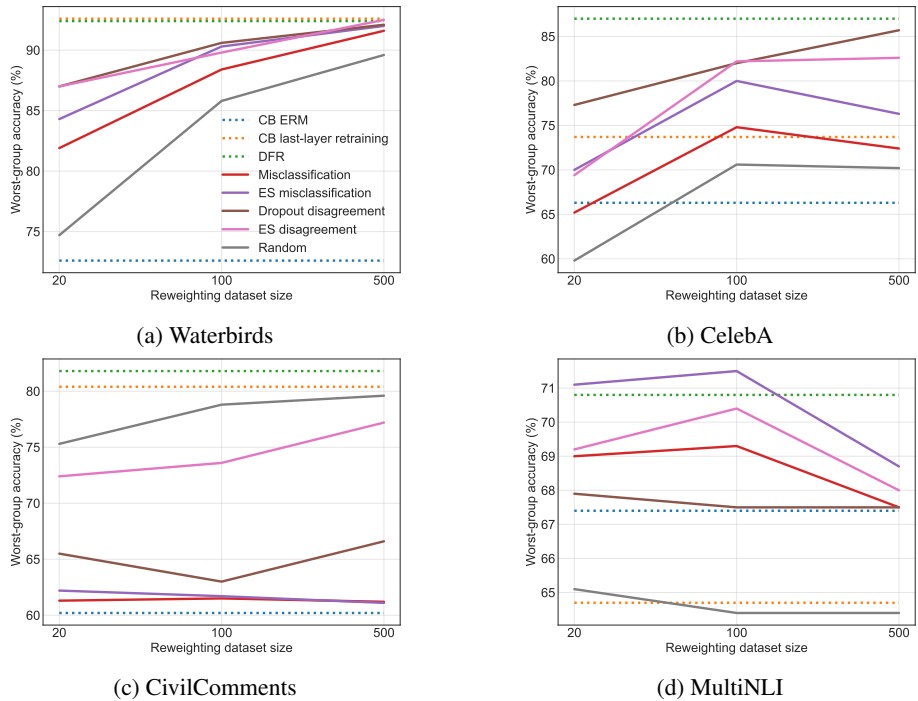

(a) Waterbirds

(b) CelebA

(c) CivilComments

(d) MultiNLI

Figure 5: **SELF reweighting dataset size ablation.** We compare the class-balanced (ERM) baseline to CB last-layer retraining, deep feature reweighting, and the four variants of SELF which we describe in Section 5. The $x$ axis is the size of the reweighting dataset, *i.e.,* the number of points selected from the held-out dataset, and the $y$ axis is the worst-group accuracy. The held-out dataset has a fixed size of 600 for Waterbirds, 9934 for CelebA, 22590 for CivilComments, and 41231 for MultiNLI. A general trend is that SELF methods tend to improve as the size $n$ of the reweighting dataset increases, except on MultiNLI. We plot the mean over three independent runs and leave out error bars for readability.

Table 9: **Total variation distance comparison.** We compare the usage of KL divergence in our early-stop disagreement SELF method to total variation distance (TVD) as used in our Theorem 1. Our method is robust to the choice of distance function (though TVD is worse on CelebA) and therefore our usage of TVD in Theorem 1 is empirically justified. We list the mean and standard deviation over three independent seeds.

| Method | Worst-group test accuracy | | | |
|---|---|---|---|---|
| | Waterbirds | CelebA | CivilComments | MultiNLI |
| KL divergence | $93.0_{\pm 0.3}$ | $83.9_{\pm 0.9}$ | $79.1_{\pm 2.1}$ | $70.7_{\pm 2.5}$ |
| Total variation distance | $92.2_{\pm 0.6}$ | $72.9_{\pm 3.4}$ | $78.6_{\pm 2.8}$ | $66.3_{\pm 2.3}$ |

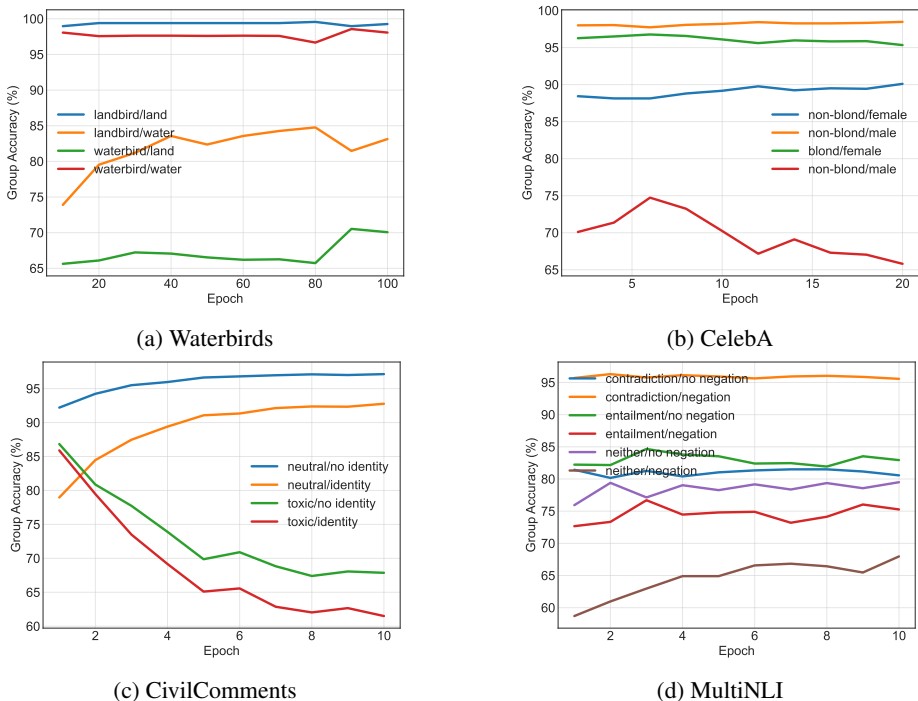

(a) Waterbirds

(b) CelebA

(c) CivilComments

(d) MultiNLI

Figure 6: **Group validation accuracies over training.** We plot the validation accuracies by group for the four benchmark datasets over the course of ERM training. Contrary to the assumptions made by JTT [41] and other early-stop misclassification methods, *the worst-group accuracy decreases with training* on CivilComments. This highlights a potential advantage of disagreement methods over misclassification methods: disagreement works regardless of whether the early-stopped or convergent model has a greater dependence on the spurious feature, as long as there is a large relative difference. We plot the mean over three independent runs and leave out error bars for readability.

Table 10: **SELF training accuracies.** We detail the training accuracies on the SELF held-out dataset using the best configuration of each method with respect to worst-group validation accuracy. We find that the misclassification techniques have much lower training accuracy than disagreement or random – as low as $0\%$ on MultiNLI – meaning that the misclassified points *cannot be fit without changing the features*. We list the mean and standard deviation over three independent seeds.

| Method | Held-out training accuracy | | | |
|---|---|---|---|---|
| | Waterbirds | CelebA | CivilComments | MultiNLI |
| Random SELF | $96.3_{\pm0.2}$ | $90.6_{\pm9.2}$ | $91.1_{\pm7.8}$ | $86.7_{\pm11.6}$ |
| Misclassification SELF | $94.6_{\pm1.6}$ | $25.3_{\pm21.9}$ | $0.4_{\pm0.2}$ | $0.2_{\pm0.3}$ |
| ES Misclassification SELF | $91.3_{\pm7.5}$ | $28.5_{\pm24.7}$ | $1.1_{\pm2.0}$ | $0.0_{\pm0.0}$ |
| Dropout Disagreement SELF | $95.5_{\pm1.4}$ | $98.7_{\pm1.5}$ | $50.9_{\pm6.3}$ | $68.3_{\pm24.7}$ |
| ES Disagreement SELF | $97.0_{\pm2.0}$ | $79.9_{\pm9.2}$ | $58.1_{\pm3.3}$ | $52.6_{\pm2.5}$ |

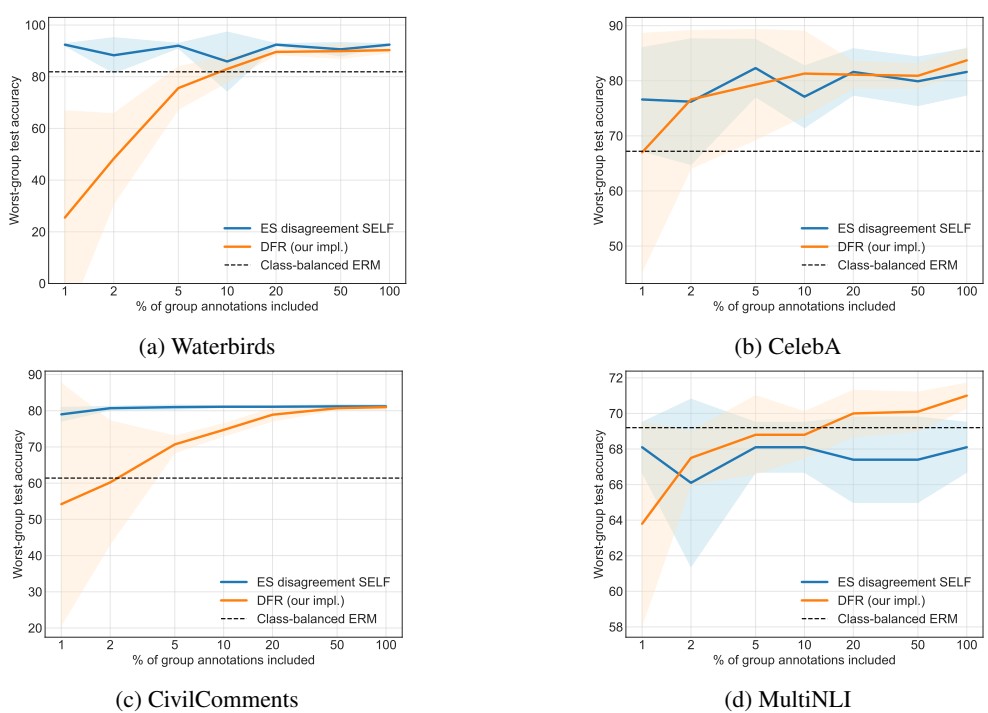

(a) Waterbirds

(b) CelebA

(c) CivilComments

(d) MultiNLI

Figure 7: **Label efficiency comparison.** We compare the performance of early-stop disagreement SELF to DFR while varying the amount of group annotations used for validation and retraining, respectively. The results show that SELF is robust to hyperparameter tuning and can massively reduce the annotation requirement: *e.g.,* at $1\%$ of data, Waterbirds has only 6 examples and CelebA has only 99 examples. Moreover, SELF retains more performance than DFR when the number of group annotations is very low. We plot the mean and standard deviation over three independent seeds.

## C Theoretical proof-of-concept for disagreement SELF

In tthis section, we describe and state our main theoretical proof-of-concept of disagreement SELF (Theorem 1). We begin with a description of our setup and assumptions.

**Setup.** Let $\mathcal{X} \subseteq \mathbb{R}^v$ be the data domain, $n$ be the sample size, and $d \geq v, n$ be the number of features. For $x \in \mathcal{X}$ and weights $\alpha, \beta > 0$ we assume that the learned predictor of both the ERM and regularized models takes the form

$$\hat{f}(x) = \alpha \phi_{\text{core}}(x) + \beta \phi_{\text{sp}}(x) + \sum_{k=1}^{d-2} \phi_{\text{junk}}^k(x) \tag{2}$$

and assume that the label $y$ is generated by $y = \text{sgn}(\phi_{\text{core}}(x))$ (*i.e.,* zero label noise). Note that this specific split of features into core, spurious, and junk categories is an extension of a common simplified setting in the literature on overparameterized linear models [82, 34], where only core and junk features are considered and only in-distribution accuracy is analyzed. We also consider the domain $\mathcal{X}$ to be partitioned into a minority group $\mathcal{X}_{\text{min}}$ and majority group $\mathcal{X}_{\text{maj}}$ such that $y\phi_{\text{sp}}(x) < 0$ for all $x \in \mathcal{X}_{\text{min}}$ and $y\phi_{\text{sp}}(x) > 0$ for all $x \in \mathcal{X}_{\text{maj}}$. (By definition, $y\phi_{\text{core}}(x) > 0$ for any $x$.)

**Assumptions on the feature learning process.** We make some extra simplifying assumptions on the feature learning process that are motivated by empirical observations in the literature. First, we assume that the learned features $\phi_{\text{core}}, \phi_{\text{sp}}, \phi_{\text{junk}}$ are *identical* for the regularized and ERM models, and are only weighted differently in the last layer. This assumption is well-justified for dropout disagreement SELF since we only use dropout for inference in the last layer (with frozen features), and for early-stop disagreement SELF since feature learning has empirically been observed to occur during the initial phase of training [29, 15]. More concretely, we consider the ERM model $\hat{f}_{\text{erm}}$ to have weights $\alpha_{\text{erm}}$ and $\beta_{\text{erm}}$, and the regularized model $\hat{f}_{\text{reg}}$ to have weights $\alpha_{\text{erm}}$ and $\beta_{\text{reg}}$. Second, we assume that weights on the core and spurious features are *normalized* such that $\alpha_{\text{erm}} + \beta_{\text{erm}} = \alpha_{\text{reg}} + \beta_{\text{reg}}$, essentially positing that the proportion of junk feature strength is the same in both ERM and regularized models. This is also justified by the empirical observation that regularization methods such as early stopping make minimal difference to in-distribution accuracy [49].

**Main result.** The following result compares a minority group test example and a majority group test example with features that are equal in magnitude, and shows that the disagreement is higher for minority group points regardless of whether the ERM or regularized model weights the spurious feature higher. We will use the *total variation distance (TVD)* as our cost function for disagreement SELF. We focus on TVD instead of KL divergence to simplify the proof; we show in Table 9 (detailed in Appendix B) that the empirical results are competitive using TVD (though worse on CelebA). We show that the difference of the TVD terms is proportional to $|\beta_{\text{erm}} - \beta_{\text{reg}}| := |\alpha_{\text{erm}} - \alpha_{\text{reg}}|$, illustrating that the models disagree more drastically the greater the discrepancy between their dependence on the spurious feature (or, equivalently, the core feature).

**Theorem 1** (Disagreement SELF). *Consider two examples $x_{\text{min}} \in \mathcal{X}_{\text{min}}$ and $x_{\text{maj}} \in \mathcal{X}_{\text{maj}}$ which have the same labels and whose features have the same magnitude, i.e., $\phi_{\text{core}}(x_{\text{min}}) = \phi_{\text{core}}(x_{\text{maj}})$, $|\phi_{\text{sp}}(x_{\text{min}})| = |\phi_{\text{sp}}(x_{\text{maj}})|$, and $\sum_{k=1}^{d-2} \phi_{\text{junk}}^k(x_{\text{min}}) = \sum_{k=1}^{d-2} \phi_{\text{junk}}^k(x_{\text{maj}})$. Let $P_{\text{min}}$ and $Q_{\text{min}}$ be distributions over $\{-1, 1\}$ which take value 1 with probability $(b(\hat{f}_{\text{erm}}(x_{\text{min}}))+1)/2$ and $(b(\hat{f}_{\text{reg}}(x_{\text{min}}))+1)/2$ respectively, and likewise for $P_{\text{maj}}$ and $Q_{\text{maj}}$.[7] Then, we have*

$$\text{TVD}(P_{\text{min}}, Q_{\text{min}}) - \text{TVD}(P_{\text{maj}}, Q_{\text{maj}}) = b \min(|\phi_{\text{core}}(x_{\text{maj}})|, |\phi_{\text{sp}}(x_{\text{maj}})|)|\beta_{\text{erm}} - \beta_{\text{reg}}| > 0. \tag{3}$$

---

[7]These are the estimated distributions we use to classify the inputs.

*Proof.* For any $x \in \mathcal{X}$ define $P(x) = (b(\hat{f}_{\text{erm}}(x)) + 1)/2$ and $Q(x) = (b(\hat{f}_{\text{reg}}(x)) + 1)/2$. By the assumptions on the features,

$$\text{TVD}(P(x), Q(x)) = |P(x) - Q(x)| \tag{4}$$

$$= \frac{b}{2}|\hat{f}_{\text{erm}}(x) - \hat{f}_{\text{reg}}(x)| \tag{5}$$

$$= \frac{b}{2}|(\alpha_{\text{erm}} - \alpha_{\text{reg}})\phi_{\text{core}}(x) - (\beta_{\text{erm}} - \beta_{\text{reg}})\phi_{\text{sp}}(x)| \tag{6}$$

$$= \frac{b}{2}|\beta_{\text{erm}} - \beta_{\text{reg}}||\phi_{\text{core}}(x) - \phi_{\text{sp}}(x)|. \tag{7}$$

Let $c = |\phi_{\text{core}}(x_{\text{maj}})|$ and $d = |\phi_{\text{sp}}(x_{\text{maj}})|$. By definition of the minority and majority groups,

$$|\phi_{\text{core}}(x_{\text{min}}) - \phi_{\text{sp}}(x_{\text{min}})| - |\phi_{\text{core}}(x_{\text{maj}}) - \phi_{\text{sp}}(x_{\text{maj}})| = c + d - |c - d| = 2\min(c, d) > 0. \tag{8}$$

Together with $b \geq 0$, Equations 7 and 8 show the result. $\qquad\square$

# D    Training details

We utilize a ResNet-50 [24] pretrained on ImageNet-1K [57] for the vision tasks and a BERT [14] model pretrained on Book Corpus [83] and English Wikipedia for the language tasks. These pretrained models are used as the initialization for ERM on the four datasets we study. We use standard ImageNet normalization with standard flip and crop data augmentation for the vision tasks and BERT tokenization for the language tasks [28]. Our implementation uses the following packages: `NumPy` [22], `PyTorch` [53], `Lightning` [72], `TorchVision` [44], `Matplotlib` [26], `Transformers` [74], and `Milkshake` [36].

For ERM and last-layer retraining (Section 4), we do not vary any hyperparameters; their fixed values are listed in Table 11. Recall that our SELF method (Section 5) splits the validation dataset in half, using the first half as a held-out dataset for selecting reweighting points and the second half for model selection [33, 28]. We vary the reweighting dataset size $n$ (the number of points selected from the held-out dataset) in the range $(20, 100, 500)$. For the vision datasets, we search over learning rates $(10^{-4}, 10^{-3}, 10^{-2})$ and train for 250 steps. For the language datasets, we search over learning rates $(10^{-6}, 10^{-5}, 10^{-4})$ and train for 500 steps. For the two early-stopping methods, we search over checkpoints saved at $10\%$, $20\%$, and $50\%$ of model training, while for dropout disagreement, we search over dropout probabilities of $0.5$, $0.7$, and $0.9$ applied only at the last layer. See Appendix B for additional ablation studies.

Table 11: **ERM and last-layer retraining hyperparameters.** We use standard hyperparameters following previous work [58, 27, 33, 28]. For last-layer retraining, we keep all hyperparameters the same except the number of epochs on CelebA, which we increase to 100.

| Dataset | Optimizer | Initial LR | LR schedule | Batch size | Weight decay | Epochs |
|---|---|---|---|---|---|---|
| Waterbirds | SGD | $3 \times 10^{-3}$ | Cosine | 32 | $1 \times 10^{-4}$ | 100 |
| CelebA | SGD | $3 \times 10^{-3}$ | Cosine | 100 | $1 \times 10^{-4}$ | 20 |
| CivilComments | AdamW [43] | $1 \times 10^{-5}$ | Linear | 16 | $1 \times 10^{-4}$ | 10 |
| MultiNLI | AdamW [43] | $1 \times 10^{-5}$ | Linear | 16 | $1 \times 10^{-4}$ | 10 |

# E  Additional tables

In this section we provide results in a tabular form to accompany figures in the main paper.

Table 12: **Table for Figure 1a.** We perform an ablation on the percentage of worst-group data used for class-balanced last-layer retraining, while keeping the total data constant. Note that the 100% numbers may not equal our other group-balanced last-layer retraining numbers due to our methodology involving downsampling and starting from a class-unbalanced ERM. We list the mean and standard deviation over three independent runs.

| Percent of worst-group data included | Worst-group test accuracy | | | |
|---|---|---|---|---|
| | Waterbirds | CelebA | CivilComments | MultiNLI |
| 2.5 | $53.5_{\pm4.4}$ | $61.1_{\pm2.9}$ | $75.3_{\pm4.9}$ | $67.3_{\pm1.8}$ |
| 5.0 | $53.5_{\pm6.0}$ | $62.8_{\pm3.9}$ | $75.4_{\pm5.0}$ | $67.4_{\pm1.9}$ |
| 12.5 | $63.1_{\pm6.3}$ | $72.0_{\pm2.8}$ | $75.5_{\pm5.1}$ | $67.6_{\pm1.9}$ |
| 25.0 | $73.6_{\pm4.2}$ | $78.2_{\pm1.7}$ | $75.5_{\pm4.9}$ | $67.8_{\pm2.1}$ |
| 37.5 | $80.2_{\pm2.9}$ | $81.5_{\pm4.2}$ | $75.6_{\pm4.8}$ | $67.8_{\pm2.1}$ |
| 50.0 | $83.2_{\pm3.7}$ | $83.7_{\pm2.8}$ | $75.8_{\pm5.0}$ | $67.9_{\pm2.0}$ |
| 62.5 | $86.2_{\pm1.3}$ | $85.7_{\pm3.1}$ | $75.8_{\pm4.8}$ | $67.9_{\pm2.0}$ |
| 75.0 | $86.5_{\pm2.1}$ | $87.0_{\pm2.5}$ | $76.1_{\pm4.7}$ | $68.0_{\pm1.9}$ |
| 87.5 | $88.8_{\pm0.6}$ | $88.2_{\pm1.3}$ | $76.0_{\pm4.7}$ | $68.3_{\pm1.8}$ |
| 100.0 | $90.4_{\pm0.9}$ | $88.7_{\pm0.7}$ | $76.1_{\pm4.6}$ | $68.3_{\pm1.9}$ |

Table 13: **Table for Figure 1b.** We compare the worst-group accuracy in Table 12 to the maximum WGA increase over a class-unbalanced ERM (see Table 5). Note that Waterbirds performs poorly at low percentages because the small size of the held-out dataset leads to no worst-group data being included as a result of our downsampling methodology. We list the mean over three independent runs.

| Percent of worst-group data included | Percent of max WGA increase over ERM | | | |
|---|---|---|---|---|
| | Waterbirds | CelebA | CivilComments | MultiNLI |
| 2.5 | $-105.0$ | 38.1 | 93.5 | $-11.1$ |
| 5.0 | $-105.0$ | 41.9 | 94.3 | 0.0 |
| 12.5 | $-51.7$ | 62.6 | 95.1 | 22.2 |
| 25.0 | 6.7 | 76.5 | 95.1 | 44.4 |
| 37.5 | 43.3 | 83.9 | 95.9 | 44.4 |
| 50.0 | 60.0 | 88.8 | 97.6 | 55.6 |
| 62.5 | 76.7 | 93.3 | 97.6 | 55.6 |
| 75.0 | 78.3 | 96.2 | 100.0 | 66.7 |
| 87.5 | 91.1 | 98.9 | 99.2 | 100.0 |
| 100.0 | 100.0 | 100.0 | 100.0 | 100.0 |

Table 14: **Table for Figure 2.** We compare class-balanced (CB) ERM on the entire dataset to splitting the dataset and performing CB ERM on the first (95%) split and CB last-layer retraining on the second (5%) split. The results with and without the held-out dataset correspond to Figure 2a and 2b respectively. We list the mean and standard deviation over three independent runs.

| Method | Held-out dataset included | Worst-group test accuracy | | | |
|---|---|---|---|---|---|
| | | Waterbirds | CelebA | CivilComments | MultiNLI |
| CB ERM | ✗ | $72.6_{\pm3.2}$ | $66.3_{\pm3.2}$ | $60.2_{\pm2.7}$ | $67.4_{\pm2.4}$ |
| CB last-layer retraining | ✗ | $72.4_{\pm3.5}$ | $74.1_{\pm2.1}$ | $80.7_{\pm1.1}$ | $64.4_{\pm1.1}$ |
| CB ERM | ✓ | $81.9_{\pm3.4}$ | $67.2_{\pm5.6}$ | $61.4_{\pm0.7}$ | $69.2_{\pm1.6}$ |
| CB last-layer retraining | ✓ | $83.1_{\pm2.7}$ | $71.3_{\pm3.2}$ | $80.8_{\pm0.2}$ | $65.6_{\pm2.4}$ |

Table 15: **Table for Figure 3.** We list the percentage of the reweighting dataset consisting of worst-group data for each SELF method. "Baseline" represents the percentage of worst-group data in the held-out dataset, while "Balanced" is the percentage of worst-group data necessary to achieve group balance (recall that there may be multiple worst groups; they are described in Section 4.1). We also include the worst-group accuracy (WGA) of each SELF method, to get a sense of whether the amount of worst-group data correlates with performance (in a loose qualitative sense). We list the mean over three independent runs and round to the nearest whole number.

| Method | WGA/Percent of worst-group data | | | |
|---|---|---|---|---|
| | Waterbirds | CelebA | CivilComments | MultiNLI |
| Baseline | 50 | 1 | 7 | 2 |
| Misclassification | 90/45 | 85/25 | 64/51 | 65/ 4 |
| ES misclassification | 91/45 | 83/ 7 | 68/37 | 68/20 |
| Dropout disagreement | 91/55 | 82/13 | 79/13 | 66/ 3 |
| ES disagreement | 93/67 | 84/ 2 | 79/24 | 71/ 4 |
| Balanced | 50 | 25 | 25 | 33 |

Table 16: **Table for Figure 4.** We compare different combinations of ERM and last-layer retraining with or without class balancing. Notably, class-balanced last-layer retraining enables nearly the same worst-group accuracy whether the ERM incorporates class-balancing or not. In Figure 4, we only plot the results which use class-balanced last layer retraining (*i.e.,* the last two rows of the table). We use the entire held-out set for last-layer retraining, and we list the mean and standard deviation over three independent runs.

| ERM class balancing | Last-layer retraining class balancing | Worst-group test accuracy | | | |
|---|---|---|---|---|---|
| | | Waterbirds | CelebA | CivilComments | MultiNLI |
| ✗ | ✗ | $87.5_{\pm 0.7}$ | $45.4_{\pm 2.3}$ | $54.7_{\pm 5.8}$ | $64.5_{\pm 0.8}$ |
| ✓ | ✗ | $88.0_{\pm 0.8}$ | $41.9_{\pm 1.4}$ | $57.6_{\pm 4.2}$ | $64.6_{\pm 1.0}$ |
| ✗ | ✓ | $92.3_{\pm 0.4}$ | $71.1_{\pm 2.4}$ | $78.4_{\pm 2.5}$ | $64.7_{\pm 1.1}$ |
| ✓ | ✓ | $92.6_{\pm 0.8}$ | $73.7_{\pm 2.8}$ | $80.4_{\pm 0.8}$ | $64.7_{\pm 1.1}$ |

# F  Broader impacts, limitations, and compute

**Broader impacts.** We hope our work contributes to the safe and equitable application of machine learning and motivates further research in ML fairness. A potential negative outcome may arise if practitioners assume their models are bias-free after applying our techniques; while we show reduced dependence on spurious correlations, no method can completely alleviate the issue, and other modes of bias may exist. We encourage practitioners to conduct rigorous examinations of model fairness.

**Limitations.** Our methods take advantage of the specificity of the spurious correlation setting, and therefore would likely underperform on datasets which exhibit a more extreme *complete correlation* (*i.e.,* contain zero minority group data) [52, 40]. Furthermore, following previous work in this setting [58, 41, 48, 33, 28], SELF utilizes a small validation dataset with group annotations for model selection. While we show in Appendix B that SELF is robust to using as few as 1% of these annotations, and our last-layer retraining method (Section 4) does not require them, completely removing this assumption is an important direction for future research.

**Compute.** Our experiments were conducted on Nvidia Tesla V100 and A5000 GPUs. We used about $25000 in compute credits in the course of this research. We believe this paper could be reproduced for under $5000 in credits, and a majority of this compute would go towards the ablation studies. With that said, our last-layer retraining methods only train the linear classifier, making them cheap and efficient to run even on older hardware.

