# OpenReview forum: "Towards Last-layer Retraining for Group Robustness with Fewer Annotations"
_NeurIPS.cc/2023/Conference — NeurIPS 2023 poster_

### Official Review · Reviewer_HP8n · 2023-06-26

**Soundness:** 2 fair
**Presentation:** 2 fair
**Contribution:** 2 fair
**Rating:** 6
**Confidence:** 3

**Summary:**

The paper tackles the aspect of last-layer retraining for better group robustness. The authors show that last-layer retraining can greatly improve worst-group accuracy with little worst-group data. Motivated by this, selective last-layer finetuning (SELF) is proposed to improve group robustness.

**Strengths:**

1. The paper focuses on the interesting aspect of improving robustness via last-layer retraining under spurious correlation. The problem is specific and practical.
2. The experiments are well done. Also, the comparisons are consistent through out the paper.
3. The findings and proposed methods are well presented. The finding of "last-layer retraining can substantially improve WGA even when the reweighting dataset has only a small proportion of worst group data" is interesting.


**Weaknesses:**

1. Novelty:

(1) Last-layer retraining/finetuning is a common techqniue used in imbalanced/long-tailed learning and various properties were found, see [1][2]. This put doubt on the novelty of the claimed finding: "holding out a subset of training data to retrain the last layer can substantially outperform ERM on the entire dataset with no additional data, annotations, or computation for training". I wonder what are the differences if we treat tail classes in [1][2] as the worst group.

(2) The proposed method: Disagreement-based methods were proposed and they were only adopted to the setting with spurious correlation. This adaptation is fine, but the performance in Table 4 cannot convince me since SELF cannot beat DFR in 2 out of 4 tasks.

2. Writing:

(1) Figure 1(a) misses legends. For Figure 1(b), please note that y-axis means relative increase to avoid confusion.

(2) The contribution part is too long.

[1] Decoupling Representation and Classifier for Long-tailed Recognition. ICLR 2020
[2] BBN: Bilateral-Branch Network with Cumulative Learning for Long-Tailed Visual Recognition. CVPR 2020

**Questions:**

Is worst-group accuracy a valid metric for evaluating robustness? A common phenomenon in long-tailed learning is that increasing tail-class accuracy often leads to decrease in head class accuracy. I recommend also adding overall accuracy as a metric for comparison. Also, WGA cannot reflect the uniformness of the overall performance.

**Limitations:**

The limitations are included in the appendix.

---

> ### Author Rebuttal · Authors · 2023-08-10
>
> We warmly thank Reviewer HP8n for their detailed comments, suggestions, and references. Below, we provide responses to each of the reviewer’s points.
>
> (Novelty 1) Thank you for the comments and the references [1, 2]; we have added citations and discussion to Section 2. We remark that the reviewer’s concern is discussed at length in the DFR paper [3] which introduced last-layer retraining to the spurious correlations setting. In their Appendix A, they argue that these methods are not directly applicable to spurious correlations robustness, and many of their points apply to our work. For example, a crucial difference is that [1] retrains on data from the training set, whereas we retrain on held-out data only. Nevertheless, we provide a worst-group accuracy comparison between the methods of [1] and our proposed methods on Waterbirds and CelebA below.
>
> | Method                   | Group Annotations | Retraining Data | Waterbirds WGA | CelebA WGA    |
> |--------------------------|-------------------|-----------------|----------------|---------------|
> | LWS [1]                  | No                | Train           | 40.0 +/- 8.1   | 35.6 +/- 14.4 |
> | cRT [1]                  | No                | Train           | 74.5 +/- 1.5   | 52.9 +/- 6.0  |
> | CB last-layer retraining (ours) | No                | Held-out        | 92.6 +/- 0.8   | 73.7 +/- 2.8  |
> | ES disagreement SELF (ours)    | No                | Held-out        | 93.0 +/- 0.3   | 83.9 +/- 0.9  |
> | DFR (our impl.)          | Yes               | Held-out        | 92.4 +/- 0.9   | 87.0 +/- 1.1  |
>
> Regarding the reviewer’s suggestion of treating tail classes as the worst group, we would like to clarify that the “worst group” is not a fixed subset of the data like tail classes are. Instead, the “worst group” is dynamic, and corresponds to the group attaining the minimum accuracy over all groups. Therefore, we cannot treat tail classes as the worst group, since we cannot explicitly control which group is the worst group. Furthermore, a “group” in our work is an element of the Cartesian product of the classes and the values of the spurious feature, so depending on the spurious feature, it may be the case that the tail classes are not all in the same group.
>
> (Novelty 2) We would like to clarify that the goal of our disagreement SELF method is **not** to beat DFR (i.e., achieve SOTA performance in the presence of group annotations). This is an unrealistic expectation, as we train with no group annotations on the reweighting dataset -- much less information than DFR, which requires group annotations for the entire reweighting dataset. Therefore, we consider DFR as an oracle method, and our goal is to get as close as possible to the oracle level. In this respect, the most important comparison to make is to other methods that also do not use group annotations, which we do in Table 1. The table shows that the performance of our methods are indeed SOTA on 3 out of 4 of the benchmarks among methods not using group annotations). Considering the exception (CelebA), we remark that the large gap to CnC on the CelebA dataset also exists in concurrent work [4], suggesting that it may be an inherent limitation of last-layer retraining and not a shortcoming of our method (since CnC modifies the features, while last-layer retraining does not). Our methods outperform all other methods with the exception of CnC on the CelebA dataset.
>
> (Writing 1) We thank the reviewer for the catch, and we have updated the figures according to the reviewer’s suggestions.
>
> (Writing 2) We thank the reviewer for the suggestion, and we have reduced the length of the contributions section by shortening the paragraphs.
>
> (Question) Thank you for the question and suggestion. We have included average accuracy numbers for ERM, class-balanced last-layer retraining, ES disagreement SELF, and DFR in Rebuttal Table 1. Our methods achieve comparable average accuracy to DFR, which is a favorable result as DFR is considered one of the SOTA methods for group robustness. We hope the average accuracy results assuage the reviewer’s concerns about how WGA alone does not reflect the uniformity of the group-accuracy distribution.
>
> [1] Kang et al. “Decoupling representation and classifier for long-tailed recognition.” ICLR 2020.
>
> [2] Zhou et al. “BBN: Bilateral-Branch Network with Cumulative Learning for Long-Tailed Visual Recognition.” CVPR 2020.
>
> [3] Kirichenko et al. “Last Layer Re-training is Sufficient for Robustness to Spurious Correlations.” ICLR 2023.
>
> [4] Qiu et al. “Simple and Fast Group Robustness by Automatic Feature Reweighting.” ICML 2023.

---

> > ### Comment · Reviewer_HP8n · 2023-08-16
> > **Update**
> >
> > Thank the authors for their detailed response. Most of my concerns are resolved and I will raise my score.

---

### Official Review · Reviewer_x5jy · 2023-07-05

**Soundness:** 3 good
**Presentation:** 3 good
**Contribution:** 2 fair
**Rating:** 3
**Confidence:** 4

**Summary:**

This paper proposes to collect a dataset from misclassifications or disagreements to fine-tune a classifier for improving sub-group accuracy in presence of spurious correlations.


**Strengths:**

The topic that this paper attempts to address is important. The paper is well-written as well. There are some interesting findings in the paper as well such as finetuning using a dataset collected from disagreements could potentially improve group-accuracy.

**Weaknesses:**

The level of novelty presented in the paper is insufficient. My primary concern stems from the method's reliance on group labels during model selection, also known as the validation set. This requirement renders these techniques nearly impractical in real-world scenarios, as defining groups is a non-trivial task even in datasets with just a hundred samples, and this becomes much more challenging for current datasets comprising billions of data points.

The Waterbirds dataset utilized in this study has been identified to contain certain bugs, which were subsequently addressed in the MaskTune paper [1]. Consequently, the current values provided may be misleading. It is worth noting that the MaskTune paper, contrary to the assumption, does not incorporate any group labels during training or model selection (validation). It would be valuable for the authors to present the worst group accuracies achieved by SELF without the utilization of any training or validation group labels. Additionally, I encourage the authors to compare these results with the methods outlined in Tables 1 and 2 of [1].

[1] Asgari, Saeid, et al. "Masktune: Mitigating spurious correlations by forcing to explore." Advances in Neural Information Processing Systems 35 (2022)


**Questions:**

Please refer to the Weaknesses section.

**Limitations:**

Please refer to the Weaknesses section.

---

> ### Author Rebuttal · Authors · 2023-08-10
>
> We warmly thank Reviewer x5jy for their comments and suggestions. Below, we provide responses to each of the reviewer’s points.
>
> (Setting) We thank the reviewer for the reference [1], and we have added a citation and discussion of [1] in Section 2. With that said, we explicitly focus on the setting where group annotations are available for model selection; this setting has been standard in the literature since JTT [2] and works in this area have contributed significant practical insights to the literature despite the annotation requirement [2, 4, 5, 6, 8]. Our main goal in this paper is **not** to beat the results in [1, 7], but rather to investigate the surprising performance of last-layer retraining in a more restrictive setting (our Section 4) and propose a SOTA algorithm for the setting of [2, 4, 5, 6, 8] based on a novel disagreement-based approach (our Section 5).
>
> (Model Selection) We have included the results of an ablation on the validation set size as Rebuttal Figure 1; the results show that ES disagreement SELF is robust to tuning hyperparameters with as little as 1% of the validation data (though variance increases as expected). Our SELF method improves greatly upon class-balanced ERM even with as few as 6 validation examples for Waterbirds and 99 examples for CelebA, massively reducing the group annotation requirement. The CivilComments and MultiNLI ablations will be done by next week and available upon request by the reviewers.
>
> (Section 4) We noticed the reviewer did not have any comments on our results in Section 4, which provided the separate insight that last-layer retraining can improve group robustness even if the reweighting dataset is drawn from the training distribution (i.e., is group-imbalanced). We would be very interested to hear your perspective on this set of results. Given the reviewer’s concern about model selection, we would like to emphasize that our results in Section 4 tune only the split ratio, which we show to be robust in Appendix D, and we recommend on Line 242 that practitioners **tune no hyperparameters at all** (therefore requiring no group annotations in the validation set). This is in contrast to concurrent work [5] whose results corroborate our Section 4.2, but they tune two required hyperparameters ($\gamma$ and $\lambda$ in their Section 3.1).
>
> (Waterbirds) Thank you for bringing the Waterbirds bugs to our attention: we have updated the paper to include a reference to this phenomenon. With that said, we believe testing on the original Waterbirds dataset is still important and useful for comparison to previous SOTA methods [2, 3, 4]. Furthermore, we test on 3 additional rigorous benchmark datasets across both vision and language domains and achieve SOTA on 2/3 of these among methods in the setting where group annotations are used for model selection but not for training.
>
> [1] Taghanaki et al. “MaskTune: Mitigating Spurious Correlations by Forcing to Explore.” NeurIPS 2022.
>
> [2] Liu et al. “Just Train Twice: Improving Group Robustness without Training Group Information.” ICML 2021.
>
> [3] Kirichenko et al. “Last Layer Retraining is Sufficient for Robustness to Spurious Correlations.” ICLR 2023.
>
> [4] Zhang et al. “Correct-N-Contrast: a Contrastive Approach for Improving Robustness to Spurious Correlations.” ICML 2022.
>
> [5] Qiu et al. “Simple and Fast Group Robustness by Automatic Feature Reweighting.” ICML 2023.
>
> [6] Sohoni et al. “BARACK: Partially Supervised Group Robustness With Guarantees”. ICML 2022 SCIS Workshop.
>
> [7] Lee et al. “Diversify and disambiguate: Learning from underspecified data.” ICLR 2023.
>
> [8] Nam et al. “Spread Spurious Attribute: Improving Worst-group Accuracy with Spurious Attribute Estimation.” ICLR 2022.

---

> > ### Comment · Reviewer_x5jy · 2023-08-18
> > **Thank you for the rebuttal**
> >
> > _"Our main goal in this paper is not to beat the results in [1, 7], but rather to investigate the surprising performance of last-layer retraining in a more restrictive setting..."_
> >
> > Given the stopping criteria for training (why stop at epochs 2, 4, 10?) and using labelled data for model selection, I don't see much novelty compared to [29], hence I keep my score.

---

### Official Review · Reviewer_Ep6f · 2023-07-10

**Soundness:** 3 good
**Presentation:** 4 excellent
**Contribution:** 3 good
**Rating:** 6
**Confidence:** 5

**Summary:**

The paper provides a detailed analysis of the DFR procedure [1]. The authors specifically consider the case when the group annotations are limited or not available during training, and show that it is still possible to largely remover the DFR performance. Specifically, the authors retrain the last layer on data with class balancing, and on data where examples are selected according to some rule (e.g. misclassified examples). The authors also provide many detailed ablations and understanding experiments that support their findings.


**Strengths:**

**S1.** The authors provide new interesting insights into the DFR procedure. For example, the results in Figure 1 on the number of minority data needed are quite surprising: even with a small increase in the proportion of minority data it is possible to significantly improve worst-group performance. Also the results in Table 3 on the effectiveness of last-layer retraining without group rebalancing are thought-provoking. Finally, the results in Figure 3, which show that better group balancing does not always lead to better WGA.

**S2.** The authors propose new practical methods (SELF), which build on DFR, but relax the requirement for group-balanced data. The results are discussed in Section 5. The authors also perform detailed ablations on the design decisions for the method, such as what cost function $c$ to use for selecting a subset of the data for last layer retraining.

**S3.** SELF can work even when the reweighting data doesn’t have class labels, and request the labels on a small subset.

Overall, I believe this paper provides new insights that expand our understanding of DFR and more generally training models robust to spurious features. The proposed methodology is also promising.


**Weaknesses:**

I believe there is one important missing experiment, and several technical details that should be explained or corrected. I explain these in detail below.

**W1.** Experiment: performance vs total number of group labeled examples

As every other group robustness method, SELF requires group-labeled data to perform hyper-parameter tuning. My understanding is that in all of the experiments, half of the validation data is used for hyper-parameter tuning. In other words, it is not true that the method can work without any group annotations (which is also true of JTT, CnC, and other methods). Moreover, SELF seems to involve more hyper-parameters than DFR (which only tunes regularization strength): length of last layer retraining, size of the reweighting dataset, learning rate, etc.

Consequently, I believe it is important to evaluate performance as a function of the total number of the group-labeled datapoints used by the method. In case of DFR, these datapoints would be used for last-layer retraining, and in case of  SELF, they will only be used for hyper-parameter tuning. I think it may be reasonable to create a group-balanced validation set of size $n$, and plot WGA vs the size $n$. For example of a similar experiment, see Fig. 5 in [2].

I believe, this experiment is quite important for evaluating the methodological contribution of the paper.

**W2.** Civilcomments

I blieve, the results presented in Table 1 for CivilComments mix two different versions of the dataset. Specifically, JTT and CnC use the [Wilds version](https://wilds.stanford.edu/datasets/#civilcomments) with 16 overlapping groups, while DFR, RWY-ES, Group-DRO and the methods described in the paper use a version with 4 groups, where all the spurious attributes are combined together. As a result, there is a big discrepancy in performance. I would recommend either (1) reporting the results for all methods on the WILDS version of the dataset, or (2) removing the JTT and CnC entries and clarifying the difference in the versions of the data.


**W3.** Waterbirds

On waterbirds, the validation data is group balanced up to class balancing. As a result, by group-balancing, the authors are able to achieve DFR performance with class-balanced last-layer retraining (CB in Table 1). However, in this case the method is virtually identical to DFR.

I think it would be more meaningful to do the experiment using a base reweighting dataset that has the same group distribution as the training data. This is the case for all other datasets.


**Questions:**

**Q1. Early stopping:** What is the early stopping criterion for the models in Section 5?

**Q2. Free lunch:** For the free-lunch results in Section 4.2 do you need regularization or early-stopping, optimized on validation WGA? Or is it quite robust to hyper-parameters?

**Q3. Hyper-parameters:** Could you please list all hyper-parameters that a practitioner needs to tune for the SELF method?

Finally, the paper [2] appears to be quite relevant, as it also attempts to automatically construct a reweighting dataset for DFR. However, [2] is a concurrent work, as it was published only after the submission of this paper. So there are no issues with the novelty of the paper.


**References**

[1] [_Last Layer Re-Training is Sufficient for Robustness to Spurious Correlations_](https://openreview.net/forum?id=Zb6c8A-Fghk); P. Kirichenko, P. Izmailov, A. G. Wilson; ICLR 2023

[2] [_Simple and Fast Group Robustness by Automatic Feature Reweighting_](https://arxiv.org/abs/2306.11074); S. Qiu, A. Potapczynski, P. Izmailov, A. G. Wilson; ICML 2023


**Limitations:**

Limitations are adequately discussed by the authors.

---

> ### Author Rebuttal · Authors · 2023-08-10
>
> We graciously thank Reviewer Ep6f for their in-depth analysis, insightful comments, and attention to detail. Below, we provide responses to each of the reviewer’s points.
>
> (W1) Thank you for this great suggestion. We have included the results of the requested ablation as Rebuttal Figure 1; the results show that ES disagreement SELF is robust to tuning hyperparameters with as little as 1% of the validation data (though variance increases as expected). Our SELF method improves greatly upon class-balanced ERM even with as few as 6 validation examples for Waterbirds and 99 examples for CelebA, massively reducing the group annotation requirement. The CivilComments and MultiNLI ablations will be done by next week and available upon request by the reviewers.
>
> We would also like to clarify that length of last-layer retraining (i.e., the number of optimizer steps) is not a hyperparameter in our method, as the number of optimizer steps is kept constant as the reweighting dataset size changes. Please see (Q3) below for additional discussion on the hyperparameters that SELF tunes.
>
> (W2) Thank you for this good catch regarding the CivilComments dataset. We have removed the CivilComments JTT and CnC entries from Table 1 and detailed the difference between the two versions of the dataset in a footnote. We would like to clarify that every method in Table 1 uses the WILDS version [1] of the dataset, but the 4-group version we study collapses all the spurious attributes (“female”, “LGBT”, etc.) into one attribute. The original (non-WILDS) CivilComments dataset [2] is not used.
>
> (W3) We thank the reviewer for the suggestion. We actually included this experiment in the paper in Figure 2(a) with the corresponding table version in Appendix C Table 7, and we remark on the reviewer’s point on Lines 229-231. By performing class-balanced last-layer retraining on the training distribution, we improve WGA by 4.8% over class-balanced ERM, which is actually larger than the 3.3% increase gained by using the validation distribution.
>
> (Q1) The early-stop criterion for SELF is the percentage of total training completed. Specifically, we use models stopped at 10%, 20%, and 50% of total training (corresponding to, e.g., 2, 4, and 10 epochs for CelebA). This procedure is detailed on lines 291-292 of the paper.
>
> (Q2) The free-lunch results in Section 4.2 only tune one hyperparameter (split ratio) and do not utilize any early stopping or regularization, except for a weight decay value of 1e-4 (not tuned). As discussed on Line 242 and Appendix D, we found that a 95/5 split worked the best in general, and we therefore recommend practitioners to tune no hyperparameters at all. Please see the (Concurrent work) section below for additional discussion about the regularization and hyperparameters used in Section 4.2.
>
> (Q3) There are three SELF hyperparameters: (1) reweighting dataset size, (2) learning rate, and (3) early-stop threshold (early-stop SELF only) or dropout probability (dropout SELF only). The number of optimizer steps is kept constant as the reweighting dataset size changes. This procedure is detailed on lines 286-293 of the paper.
>
> (Concurrent work) We thank the reviewer for bringing up the reference [3], which we had also noticed after the initial submission of our work. We believe their results corroborate our findings in Section 4.2. A key difference between our methods is that we only tune one hyperparameter (split ratio) in Section 4, while [3] includes two hyperparameters ($\gamma$ and $\lambda$ in their Section 3.1) which tune mistake upweighting and $\ell_2$ regularization towards the original last-layer weights. Our ablation in Appendix D shows that our Section 4 results are robust to split ratio and we recommend on Line 242 that practitioners **tune no hyperparameters at all**; therefore, our results are more practical because we do not require group annotations in the validation set. We remark that our Section 4.1 and Section 5 are entirely novel and no corresponding results appear in [3]. We have added a citation and discussion of [3] to Section 2.
>
> [1] Koh et al. “WILDS: A Benchmark of in-the-Wild Distribution Shifts.” ICML 2021.
>
> [2] Borkan et al. “Nuanced metrics for measuring unintended bias with real data for text classification.” WWW 2019.
>
> [3] Qiu et al. “Simple and Fast Group Robustness by Automatic Feature Reweighting.” ICML 2023.

---

> > ### Comment · Reviewer_Ep6f · 2023-08-21
> > **Thank you for the rebuttal**
> >
> > Dear authors, thank you for the rebuttal and clarifications. I am mostly satisfied with the response. For the experiment on the dependence on the number of labeled validation points, it would be great to see a comparison with DFR specifically, to make sure that the proposed method improves label efficiency compared to DFR. That being said, I believe this paper provides a nice contribution,  and I vote for acceptance.

---

> > > ### Author Response · Authors · 2023-08-21
> > > **Requested experiments**
> > >
> > > Thank you for your feedback and continued engagement. The requested experiments finished during the discussion period, so we have included them below (except for CivilComments, which is still running). The results show that ES disagreement SELF has comparable or better group annotation efficiency than DFR, particularly at 1-2% of data. Compared to Figure 5 in Qiu et al, our SELF method displays similar scaling behavior to AFR.
> > >
> > > ***Waterbirds***
> > >
> > > | Method/Group Annotations | 1%            | 2%            | 5%           | 10%           | 20%          | 50%          | 100%         |
> > > |--------------------------|---------------|---------------|--------------|---------------|--------------|--------------|--------------|
> > > | DFR (our impl.)          | 25.5 +/- 41.3 | 48.3 +/- 17.4 | 75.6 +/- 8.2 | 83.0 +/- 5.1  | 89.6 +/- 1.0 | 89.9 +/- 2.8 | 90.3 +/- 1.1 |
> > > | ES Disagreement SELF     |  92.4 +/- 0.4 |  88.3 +/- 6.8 | 92.0 +/- 0.9 | 85.9 +/- 11.4 | 92.4 +/- 0.4 | 90.6 +/- 2.6 | 92.4 +/- 0.4 |
> > >
> > > ***CelebA***
> > > | Method/Group Annotations | 1%            | 2%            | 5%            | 10%          | 20%          | 50%          | 100%         |
> > > |--------------------------|---------------|---------------|---------------|--------------|--------------|--------------|--------------|
> > > | DFR (our impl.)          | 67.0 +/- 21.6 | 76.6 +/- 12.5 | 79.3 +/- 10.0 | 81.3 +/- 7.7 | 81.1 +/- 2.4 | 80.9 +/- 2.2 | 83.7 +/- 2.3 |
> > > | ES Disagreement SELF     |  76.6 +/- 9.4 | 76.2 +/- 11.4 | 82.3 +/- 5.2  | 77.1 +/- 5.6 | 81.6 +/- 4.2 | 79.9 +/- 4.4 | 81.6 +/- 4.2 |
> > >
> > > ***MultiNLI***
> > > | Method/Group Annotations | 1%           | 2%           | 5%           | 10%          | 20%          | 50%          | 100%         |
> > > |--------------------------|--------------|--------------|--------------|--------------|--------------|--------------|--------------|
> > > | DFR (our impl.)          | 63.8 +/- 5.7 | 67.5 +/- 1.5 | 68.8 +/- 2.2 | 68.8 +/- 1.3 | 70.0 +/- 1.3 | 70.1 +/- 1.1 | 71.0 +/- 0.7 |
> > > | ES Disagreement SELF     | 68.1 +/- 1.4 | 66.1 +/- 4.7 | 68.1 +/- 1.4 | 68.1 +/- 1.4 | 67.4 +/- 2.4 | 67.4 +/- 2.4 | 68.1 +/- 1.4 |

---

### Official Review · Reviewer_KCny · 2023-08-05

**Soundness:** 3 good
**Presentation:** 3 good
**Contribution:** 3 good
**Rating:** 7
**Confidence:** 4

**Summary:**

This work tackles an important problem of preventing the reliance of neural networks on spurious correlations. It builds on top of the work [1] primarily by using last layer re-training on class balanced held out dataset without the need for group annotations. They also additionally propose a simple but effective method SELF wherein the samples in disagreement with regularized models in the prediction with class annotations can be used for re-training the last layer. This method is particularly useful when group imbalance is large.
Results are shown on Waterbirds, CelebA, MultiNLI and CivilComments datasets and performance is comparable to DFR [1].

**Strengths:**

(1) The paper is well written and does very good analysis with ablation studies.

(2) Makes an important observation that it is ok if re-weighting dataset has small proportion of worst group data. This proves that the class balancing is more important than the group balancing.

(3) The paper addresses some of the practical challenges (in terms of computation and training overhead) by eliminating need to annotated group labels which is expensive, need to train twice, need to train the original ERM on class-balanced dataset.

(4)The provided solutions are simple, and will not introduce any implementation or annotation overhead for already trained large scale models.
The paper addresses the limitation of accuracy gap when the group imbalance is high and proposes a solution for the same with SELF technique.

**Weaknesses:**

(1)The observation of class-balancing importance and re-evaluation of group balancing’s role in spurious correlation is a valuable contribution. However, solution in itself lacks novelty as it is not proposing a new algorithm per se. It seems very similar to [1] "Last layer re-training is sufficient for robustness to spurious correlations"  with lesser constraints in terms of re-training dataset.

(2) This particular line under section 5  “In addition to the balance of the reweighting, dataset, it is likely that characteristics of the specific data selected also contribute to SELF results. “ is unclear.  Can you provide some qualitative examples from Celeb A where drop-out SELF disagreement does better than misclassification. More insights on what aspects of data makes the dropout SELF better would be good. Similarly, more qualitative example on the “uncertain data points” selected by the SELF-disagreement would provide more valuable insights. A lot of the observations are not supported by reasoning.

(3) Since the key claim is the advantage of not needing group labels annotations - ablation studies on what is the overhead caused for obtaining group annotations (probably annotation time) will help quantify the effectiveness of the proposed method.
Similarly ablation studies on the training compute time / mem consumotion between [1] and your method can help quantify the gains if any in training ERM on 95% split and using a MLP instead of Logistic regression.


References:
[1] "Last layer re-training is sufficient for robustness to spurious correlations"

**Questions:**

Covered in weaknesses.

**Limitations:**

A lot of the questions posed by the paper are still not addressed or analyzed . For e.g., It would be great to see more intuitive and theoretical reasoning on fundamental observations like: why last layer re-training and retraining on held out small class balanced data is improving the group robustness. Though experimental evidence has been provided, further analysis on the theoretical reasoning behind this would make a significant contribution in this subject.

---

> ### Author Rebuttal · Authors · 2023-08-10
>
> We graciously thank Reviewer KCny for their insightful comments and thoughtful suggestions. Below, we provide responses to each of the reviewer’s points.
>
> (Weakness 1) We would like to clarify that we do not claim algorithmic novelty for the results in Section 4, only for our disagreement SELF algorithm in Section 5. We believe that our contributions in Section 4 instead fall in the category of foundational empirical insight; our experiments reveal previously unknown phenomena which are surprising in the context of previous literature and have important ramifications for future algorithm design. In particular, our Section 4 results call into question **why** DFR improves worst-group accuracy, as we show that perfect group balance is not needed to achieve good performance, and that last-layer retraining on the training distribution can be surprisingly effective. We hope these insights will motivate future theoretical work to understand the underlying foundations of spurious correlations, as well as motivate future empirical work that will not only push the boundaries of SOTA worst-group accuracy, but also provide simpler and more interpretable algorithms.
>
> With that said, our disagreement SELF algorithm (Section 5) is entirely novel and differs from previous work [2, 3] in several important ways (besides not requiring group annotations). First, SELF finetunes the last layer instead of retraining as in [2]. Second, we show that tuning the $\ell_1$ regularization as in [2] is unnecessary and good performance can be achieved by fixed $\ell_2$ regularization, removing a hyperparameter. Third, we show that the best performance is achieved when using disagreements instead of misclassifications as in [3]. Fourth, while [3] assumes that the early-stopped model has low worst-group accuracy (causing misclassifications), SELF works even when the early-stopped model has higher worst-group accuracy than the convergent model (this is the case on CivilComments).
>
> (Weakness 2) We have included qualitative examples for misclassification SELF, dropout disagreement SELF, and early-stop disagreement SELF as Rebuttal Figure 2. With that said, we caution against drawing conclusions from a handful of qualitative examples. Therefore, we appeal to mathematical theory to justify why disagreement works well, and we recently proved a relevant theorem described in the (Limitations) section below. Likewise, we believe our observations on the distinction between misclassification and disagreement points are well-supported by previous study in the theory literature, including the references we cite on Lines 310-313.
>
> (Weakness 3) While our methods remove the need to annotate thousands of examples, we believe our most significant contribution is in enabling new capabilities (e.g., in settings with strict privacy or fairness constraints, disallowing or reducing the very presence of annotations) rather than accelerating existing annotation pipelines. For example, racial and gender identity are among the most important and socially relevant spurious features, but these attributes are sensitive and often cannot be collected in the first place (making annotation time irrelevant). One could imagine a scenario where a limited number of users may voluntarily self-identify, but too few to run DRO or even DFR; in that case, annotation time is negligible and the labeled samples may be used for model selection in our SELF algorithm. The reviewer may also be interested to see the results of an ablation on the validation set size, which we included as Rebuttal Figure 1.
>
> We did not perform a detailed ablation study on the time/memory consumption of SGD vs logistic regression since the compute required is negligible on modern hardware regardless of implementation (roughly 2-4MiB of RAM and 10-20 minutes of training for one SELF instance on one Nvidia V100 GPU). With that said, the previous logistic regression version must pre-compute and save to disk the feature embeddings for the entire dataset, which can take up several GB and can be slow depending on disk I/O speed (e.g., if using an HDD rather than an SSD).
>
> (Limitations) We agree wholeheartedly with the reviewer that theoretical justification for the empirical phenomena identified in this paper is interesting and important. To this end, we have been considering multiple theoretical questions since the submission. (1) As the reviewer suggested, an intriguing question is why class-balanced last-layer retraining improves worst-group accuracy without group annotations. We have considered several possible avenues here, including whether last-layer retraining has a sparsity-inducing implicit regularization effect and whether it uses the held-out data to learn a non-degenerate classifier after the training data experiences neural collapse [1]. With that said, we think this question is interesting and difficult enough to be a separate paper of its own. (2) We are also interested in theoretical justification for the disagreement-based upsampling of minority groups phenomenon observed in Figure 3. On this front, we recently proved a result which shows that model disagreement provably upsamples minority group points. In particular, for an overparameterized linear regression setting with frozen features and four groups, we show that the KL divergence between the regularized and convergent models is always higher for minority group points than majority group points.
>
> [1] Papyan et al. “Prevalence of Neural Collapse during the terminal phase of deep learning training.” PNAS 117 (40) 24652-24663.
>
> [2] Kirichenko et al. “Last Layer Retraining is Sufficient for Robustness to Spurious Correlations.” ICLR 2023.
>
> [3] Liu et al. “Just Train Twice: Improving Group Robustness without Training Group Information.” ICML 2021.

---

### Author Rebuttal · Authors · 2023-08-10

Please see the attached PDF file for additional figures and tables.

---

### Decision · Program_Chairs · 2023-09-21

**Decision:**

Accept (poster)

**Comment:**

This paper received diverging reviews. Reviewer x5jy expressed concern regarding novelty whereas the rest 3 reviewers acknowledge the contribution of this work. Upon reading through the paper and all the discussions, the AC feels that this work does contain enough novelty/contributions to be accepted despite not being groundbreaking, given the relaxed requirements on the annotations on held-out dataset. In light of this consideration, the AC recommends acceptance.